# Rapidly damping hydrogels engineered through molecular friction

Zhengyu Xu[1,2,6], Jiajun Lu[1,6], Di Lu[3], Yiran Li[1], Hai Lei[4], Bin Chen[3], Wenfei Li[1,2], Bin Xue[1] ✉, Yi Cao[1,2,5] ✉ & Wei Wang[1,2] ✉

Hydrogels capable of swift mechanical energy dissipation hold promise for a range of applications including impact protection, shock absorption, and enhanced damage resistance. Traditional energy absorption in such materials typically relies on viscoelastic mechanisms, involving sacrificial bond breakage, yet often suffers from prolonged recovery times. Here, we introduce a hydrogel designed for friction-based damping. This hydrogel features an internal structure that facilitates the motion of a chain walker within its network, effectively dissipating mechanical stress. The hydrogel network architecture allows for rapid restoration of its damping capacity, often within seconds, ensuring swift material recovery post-deformation. We further demonstrate that this hydrogel can significantly shield encapsulated cells from mechanical trauma under repetitive compression, owing to its proficient energy damping and rapid rebound characteristics. Therefore, this hydrogel has potential for dynamic load applications like artificial muscles and synthetic cartilage, expanding the use of hydrogel dampers in biomechanics and related areas.

Hydrogels capable of withstanding various deformations and efficiently dissipating energy during dynamic loading and unloading cycles are increasingly demanded for wearable sensors, soft robotics, and tissue engineering[1–7]. The toughening and energy dissipation in hydrogels are often achieved through viscoelastic dissipation mechanisms[8–10], such as sacrificial bonds or networks[11–15], involving hydrophobic interactions[16–18], ionic pairing[19–21], hydrogen bonding[22–25], coordination interactions[26–30], host–guest interactions[31–33], and microcrystals[34,35]. The breaking of sacrificial bonds or interactions results in the release of both the enthalpic energy stored within these bonds and the elastic energy stored in the polymer chains connected by these bonds. This process leads to an increase in hysteresis and the apparent work needed to disrupt the internal network. However, to enhance energy dissipation, the energy of these sacrificial bonds needs to be substantial, which can lead to generally slow reformation

rates[36,37]. The slow reformation and the inability to find the original pair in hydrogel networks compromise the fast recovery of hydrogels[27,38–40]. While the topological recovery of networks driven by elasticity is fast, the reformation of sacrificial bonds tends to be slower, leading to less efficient recovery of the hydrogel's energy dissipation capacity. This discrepancy poses a challenge in synthesizing hydrogels that can simultaneously achieve rapid recovery and effective energy dissipation.

Friction, alongside sacrificial bonds/networks, plays a crucial role in energy dissipation in nature[41–43]. At the atomic and molecular levels, internal or intermolecular friction effectively dissipates energy through sliding mechanisms and forces perpendicular to these directions[44–48]. Consequently, tuning the normal force and sliding velocity can adjust energy dissipation. We suggest that this molecular-level friction can also enhance bulk-level energy dissipation in

[1]Collaborative Innovation Center of Advanced Microstructures, National Laboratory of Solid State Microstructure, Department of Physics, Nanjing University, Nanjing 210093, China. [2]Institute for Brain Sciences, Nanjing University, Nanjing 210093, China. [3]Department of Engineering Mechanics, Zhejiang University, Hangzhou 310027, China. [4]School of Physics, Zhejiang University, Hangzhou 310027, China. [5]Chemistry and Biomedicine innovation center, Nanjing University, Nanjing 210093, China. [6]These authors contributed equally: Zhengyu Xu, Jiajun Lu. ✉e-mail: xuebinnju@nju.edu.cn; caoyi@nju.edu.cn; wangwei@nju.edu.cn

hydrogels. In contrast to the time-dependent recovery associated with energy dissipation through sacrificial bond rupture or network disentanglement, friction-based energy dissipation in hydrogels is active only during deformation and halts immediately after load release, thus minimally affecting the network structure. Additionally, the swift molecular restoration following friction significantly enhances the recovery rate of the hydrogels.

Friction structures are commonly found in many slide-ring hydrogels, where polyethylene glycol (PEG) chains are interlinked by cyclodextrin (CD) rings acting as slidable crosslinks[31,49-55]. In these hydrogels, CD rings threaded onto PEG chains serve as crosslinkers. The network's properties can be optimized by adjusting the location of these rings, enhancing uniformity during stretching or compression. This uniformity allows the hydrogels to bear mechanical loads more efficiently and prevent force concentration, making slide-ring hydrogels notably stretchable and tough[31,49,51]. In these slide-ring hydrogels, polymer chains can pass through CD rings, which act like pulleys to cooperatively equalize the tension of polymer chains[52,55]. The tension equalization occurs not only within individual polymer chains but also among adjacent interlocked cross-links. This pulley effect results in CD rings being positioned where tension on both sides is balanced after the initial pulling cycle, limiting their subsequent movement.

In this study, we demonstrate that using a short polymer-crosslinked CD ring dimers as slidable chain walkers can achieve efficient chain friction. This friction occurs as the chain walkers slide along the hydrogel network during deformation. The hydrogels, made of covalently crosslinked four-armed PEG, ensure that network recovery is independent of CD ring movement. The polymer strands between covalent crosslinks define the movement range of CD rings during network deformation and facilitate their efficient return to original positions after load release. This structure guarantees effective mechanical damping and rapid recovery.

## Results

### Molecular and network engineering of hydrogels containing chain walkers

In conventional slide-ring hydrogels, CD rings threaded onto PEG chains act as mobile crosslinkers. This design allows the rings to shift during network reconfiguration, maintaining PEG crosslinking and imparting the hydrogel with notable stretchability (illustrated in Fig. 1a, c). Considering the free energy of the hydrogel network, the initial state (i of Fig. 1a, b) is metastable due to the random distribution of CD rings. When stretched, the network's uniformity increases as the rings adjust to maximize extension, causing the free energy to reach its peak (ii of Fig. 1a, b). However, after relaxation, the network reverts to a more uniform arrangement with the CD rings distributed more evenly, resulting in a energy level which is slightly different from that in the initial state (iii of Fig. 1a, b). Due to the high energy barrier ($E_1$), the CD rings might not return to their original position while the hydrogel can recover its original shape (Fig. 1a, c). Consequently, the CD rings cannot undergo repeated and long-range frictions during the deformation cycles, resulting in minimal hysteresis and energy dissipation in these slide-ring hydrogels after the first round of cyclic deformation[31,52,56].

We propose a method that leverages molecular frictions between CD rings and PEG chains (as illustrated in Fig. 1d–f). In this approach, CD rings are linked by a short PEG segment (CD-PEG-CD), functioning as chain walkers on longer PEG strands within the hydrogel network, where the molecular weight of the PEG strands is five folds that of the linkers (Fig. 1d, f). This design allows the CD rings to experience normal forces during deformation. Unlike traditional designs where rings slide on free-ended polymers, our design secures both ends of the PEG strands to covalent crosslinks, creating a railway system (i of Fig. 1d). When subjected to load, this railway strand stretches, enabling the CD rings to slide along it ($P_1 \rightarrow P_1'$ and $P_2 \rightarrow P_2'$) under the stretching force, thereby dissipating energy through molecular friction effectively (ii of

Fig. 1d). Thanks to the elasticity of the covalently crosslinked hydrogel network, the railway strands return to their original position upon the release of the load (iii of Fig. 1d). During the relaxation of the hydrogel, the chain walkers (CD-PEG-CD) remain stationary when friction-based forces prevent diffusion. However, as the tetra-arm PEG network undergoes entropic recoiling (depicted as the railway in Fig. 1d), it draws the CD rings closer together, notably reducing their separation compared to the original distance ($P_1'P_2'$ is smaller than $P_1P_2$). At this point, the PEG segment linking CD rings relaxes with the restoration of the tetra-arm PEG network, alleviating any stretching force applied to the CD rings. Consequently, the chain walkers (CD-PEG-CD) can rapidly return to their original positions via free diffusion, bypassing the challenge of overcoming high friction forces encountered under elevated nominal forces. Additionally, the energy barrier between states i and iii in the case of Fig. 1e ($E_2$) is smaller than that of Fig. 1b ($E_1$), facilitating the rapid recovery rate of chain walkers under thermal fluctuation. Noting that two CD rings of a chain walker may attach to the same PEG chain during gelation and act as the intra-chain slide-ring linker, which would not contribute to the friction-based energy dissipation of hydrogels.

### Frictions of chain walkers at the molecular level

The molecular-level friction between PEG chains and CD rings (α-CD and β-CD) was explored using single molecule force spectroscopy (SMFS) with an atomic force microscope[57-60]. In this setup, as shown in Fig. 2a, a PEG chain (10 kDa) was attached to a silicon nitride cantilever tip, while amino-modified CD was covalently linked to glass substrates. As the cantilever approached the substrate, the PEG chain could possibly thread through the CD ring. In about 1.65% of the trials, we could successfully thread the PEG chain to the CD ring. In those cases, the PEG chain was pulled out from the CD ring during retraction, recording the friction between them (Fig. 2a and Supplementary Fig. 1). The representative SMFS traces for these events showed distinct force plateaus at several dozen piconewtons, characterizing the friction between PEG and CD (Fig. 2b). These rupture distances were all less than 80 nm, aligning with the maximum length of the PEG chain[61]. In addition, in the cases the PEG chain failed to thread through the CD ring, the plateaus were absent in the SMFS traces. Control experiments on unmodified substrates showed significantly reduced pickup rates, further confirming that the observed force plateaus were due to PEG-CD friction (Supplementary Fig. 2). Moreover, no obvious difference between pickup rates of α-CD and β-CD was observed, indicating the similar success rates of threading a PEG chain through different CD rings in SMFS experiments. The measured frictional forces between PEG and α-CD or β-CD at a pulling speed of 200 nm s⁻¹ were 26 and 19 pN, respectively, as shown in Fig. 2c. α-CD exhibited a slightly higher frictional force than β-CD, likely due to its smaller inner diameter. Dynamic force spectroscopy further assessed this friction (Fig. 2d, e), revealing that α-CD's frictional force exceeded β-CD's across various pulling speeds (200 to 3200 nm s⁻¹). Notably, the frictional forces tripled when the pulling speed increased from 200 to 3200 nm s⁻¹, suggesting that CD-PEG friction could be considered intermolecular interactions. This finding implies that the energy dissipation from molecular friction escalates with increasing deformation rates. In single molecule force spectroscopy (SMFS), the friction force is counteracted by the tethering force of the linker connecting to the CD ring, resulting in a constant force plateau in the force-extension curves. The greater the tethering force, the greater the friction force (Fig. 2d, e). We can anticipate that the friction force becomes insignificant if the CD ring diffuses freely on a PEG molecule without the tethering forces exerted by the linker linked to the CD ring, akin to when the pulling speed approaches zero. Moreover, by using the Bell-Evans model[62] to extrapolate to zero pulling force, and assuming the diffusion process involves the CD jumping within a periodic potential dictated by the internal interaction between CD and the PEG chain[63], we determined

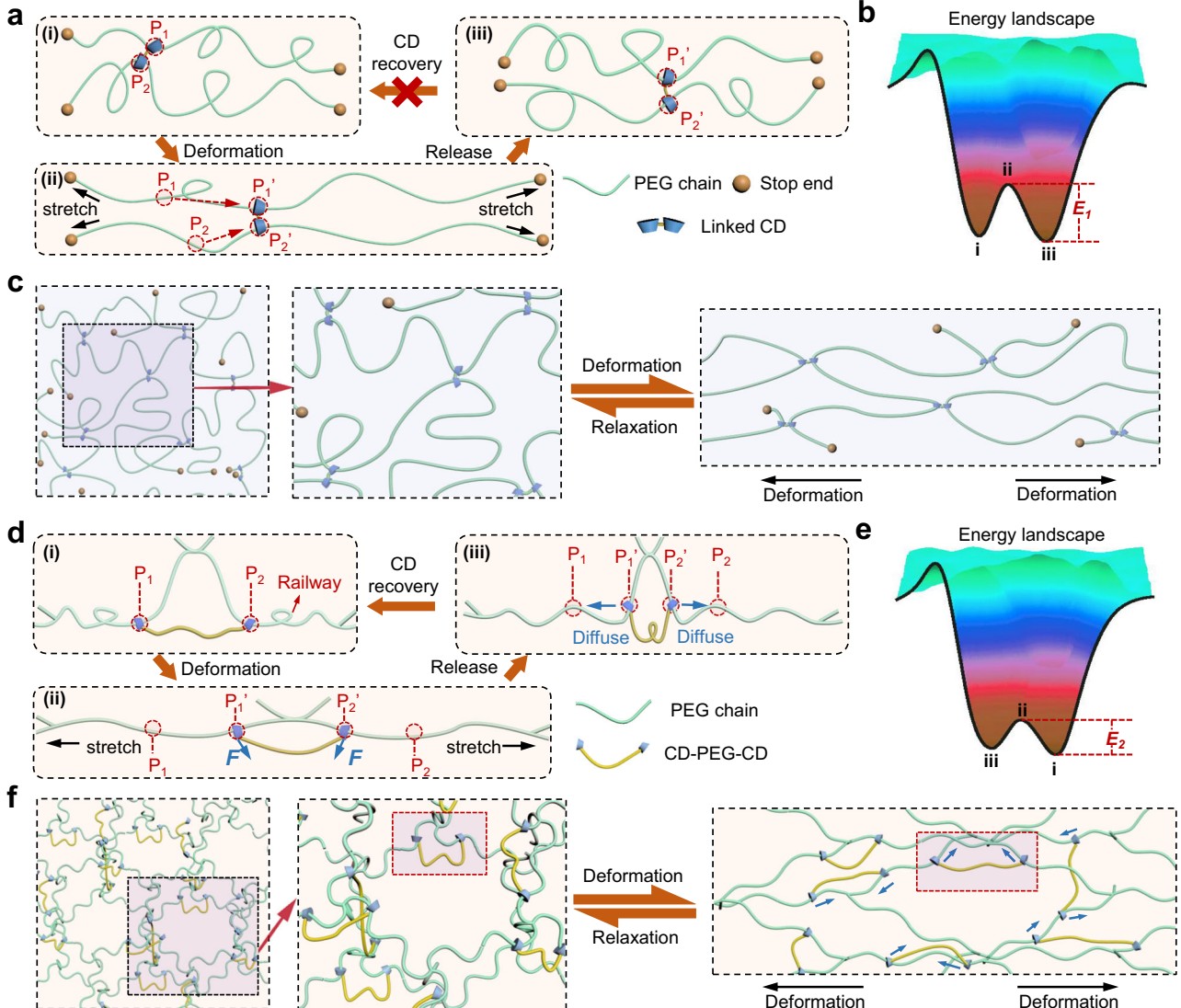

**Fig. 1 | Molecular and network engineering of hydrogels containing chain walkers. a** Schematic of CD sliding on end-free PEG chains in conventional slide-ring hydrogels. CD slides along the PEG chain and barely restores after the stress relaxation. $P_1$, $P_2$, $P_1'$ and $P_2'$ indicate fixed positions of the polymer. The × mark between state i and state iii indicates that the CD rings might not be able to return to their original position. **b** Schematic free energy landscape corresponding to different states in **a**. $E_1$ corresponds to the energy barrier between state ii and state iii. **c** Schematic of the polymer network in conventional slide-ring hydrogels under deformation and relaxation. **d** Schematic of CD sliding on end fixed PEG chains in

hydrogels of this work. CD sliding can dissipate energy efficiently, and the position of the chain walkers can be rapidly restored via diffusion after the stress relaxation. $P_1$, $P_2$, $P_1'$ and $P_2'$ indicate fixed positions of the railway polymer. **e** Schematic free energy landscape corresponding to different states in **d**. $E_2$ corresponds to the energy barrier between state i and state ii. **f** Schematic of the hydrogel network made of covalently crosslinked four-armed PEG containing chain walkers under deformation and relaxation. CD slides and restores along PEG chains under dynamic load-bearing cycles. Blue arrows indicate the sliding directions of chain walkers along railway polymers.

the diffusion speed at zero pulling forces to be 23 nm s$^{-1}$ for α-CD and 26 nm s$^{-1}$ for β-CD (See Supporting Information for the calculation details). This suggests that CD rings can diffuse rapidly in the absence of pulling forces, indicating that CD rings can overcome frictional forces for rapid diffusion when no external forces are applied to them.

Molecular dynamics (MD) simulations provided further insight into the sliding friction and molecular interactions (Fig. 2f and Supplementary Fig. 3). The simulations involved pulling a PEG chain through a CD ring and then in water while recording the work done by the pulling forces (top of Fig. 2g). The force exerted was deduced from the slopes of each linear region, revealing a distinct force plateau when PEG was pulled through CD (bottom of Fig. 2g). The discrepancy in pulling forces between CD and water was attributed to friction forces (Supplementary Fig. 4). This friction forces doubled when the pulling speed increased from 1 to 14 m s$^{-1}$ (Fig. 2h, i), aligning with the trends

observed in the SMFS experiments. The simulations also showed slightly lower friction forces between PEG and β-CD compared to α-CD. Noting that the rupture force ($F$) is positively correlated to the pulling speeds according to the Bell-Evans model[58,62,64], in which the correlation between the most probable rupture force ($F$) and loading rate ($r$) can be described as $F = \frac{k_B T}{\Delta x} \ln(\frac{\Delta x}{k_{off} k_B T}) + \frac{k_B T}{\Delta x} \ln(r)$. As a result, the calculated frictional forces in MD simulations were much higher than those in SMFS experiments due to the much higher pulling speed. Additionally, significant fluctuations in the number of accumulated hydrogen bonds were detected as PEG was pulled (Supplementary Fig. 5a). In a more detailed analysis, the number of hydrogen bonds between a fixed oxygen atom on PEG and the α-CD ring oscillated between 0 and 1 as the pulling distance increased, suggesting the formation and rupture of hydrogen bonds in a stepwise manner

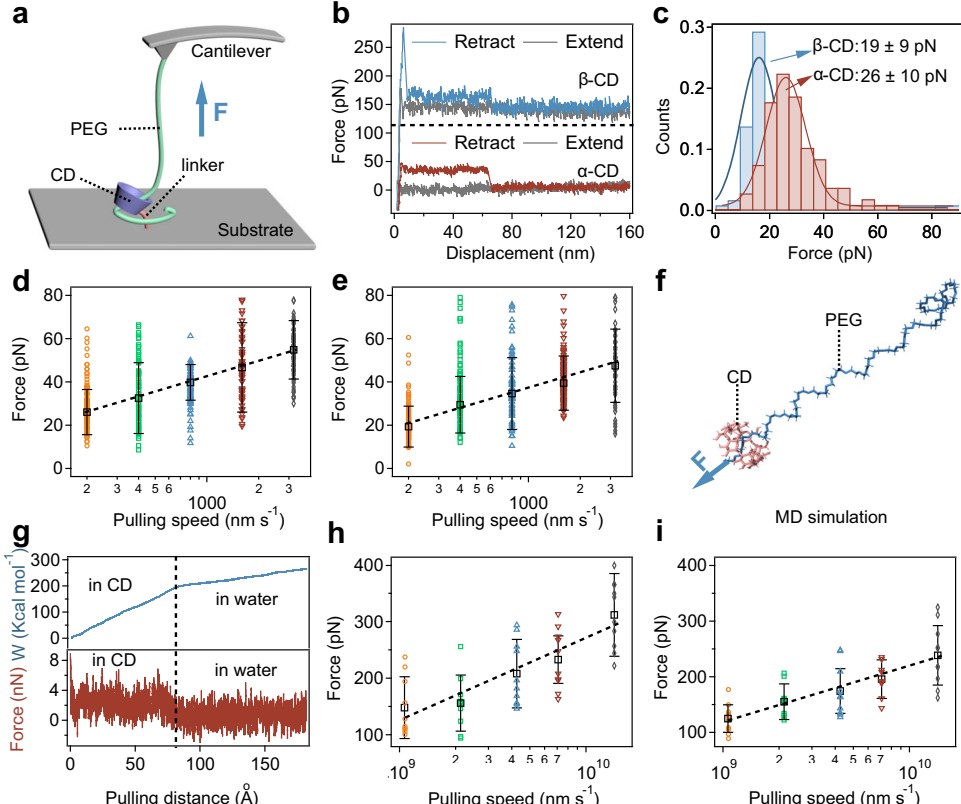

**Fig. 2 | SMFS and MD simulations of the friction between PEG and CD.**
**a** Schematic diagram of the AFM-based SMFS experiments of the friction between PEG and CD. mPEG-SH (10 kDa) was linked to the cantilever tip via APTES and SMCC. **F** indicates the pulling force. **b** Typical force–displacement curves for the friction between PEG and α-CD or β-CD at a pulling speed of 200 nm s⁻¹. **c** Force histograms of frictions between PEG and α-CD or β-CD at a pulling speed of 200 nm s⁻¹. The Gaussian fitting shows average frictional forces of 26 ± 10 pN ($n = 107$) and 19 ± 9 pN ($n = 122$), respectively. **d**, **e** Dynamic force spectroscopy experiments of sliding frictions between PEG and α-CD (**d**) or β-CD (**e**) at various pulling speeds (200, 400, 800, 1600, and 3200 nm s⁻¹). Frictional forces between PEG and α-CD or β-CD are both pulling speed dependent. The dashed line indicates the linear fitting of the frictional forces and pulling speeds. The diffusion speeds at

zero pulling forces were determined to be 23 nm s⁻¹ for α-CD and 26 nm s⁻¹ for β-CD. Values represent the mean and standard deviation (sample size $n = 107, 76, 77, 60$ and 81 in **d**, respectively; sample size $n = 122, 86, 75, 140$, and 62 in **e**, respectively). **f** Cartoon representation of pulling a PEG ($n = 34$) through the ring of CD in a typical MD simulation. All the force fields of molecules were generated based on the general AMBER force field (GAFF) and Antechamber. **F** indicates the pulling force. **g** Typical curves of work vs pulling distance (top) and force vs pulling distance (bottom) in α-CD and water. PEG was first pulled through a α-CD ring and then pulled in water. **h**, **i** Frictional force between PEG and α-CD (**h**) or β-CD (**i**) at different pulling speeds (1–14 m s⁻¹) predicted by MD simulations. Values represent the mean and standard deviation (sample size $n = 10$ for the frictional forces at each pulling speed).

(Supplementary Fig. 5b). These MD simulation results not only corroborate the existence of frictional force between PEG and α-CD but also suggest that this friction is largely due to hydrogen bonding, explaining the variation in friction forces at different pulling speeds observed in the dynamic force spectroscopy.

### Hydrogel preparation and mechanical properties

Next, we explored if molecular friction from chain walkers (CD rings connected by short PEG linkers) could influence the energy dissipation of hydrogels at the macroscopic level. A series of hydrogels, featuring these chain walkers (CD-PEG-CD) incorporated into the hydrogel network (Fig. 1f), were synthesized using maleimide-terminated 4-armed PEG (20 kDa, PEG-Mal) and thiol-terminated 4-armed PEG (20 kDa, PEG-SH). We synthesized α-CD or β-CD dimers tethered by a short PEG linker (2 kDa) as chain walkers (named as PEG-α-CD and PEG-β-CD, respectively, Supplementary Fig. 6a). Analysis by ¹H NMR and UV spectroscopy confirmed ~80% conjugation of PEG and CD (Supplementary Figs. 6 and 7).

In a typical hydrogel preparation, PEG-CD and PEG-SH were premixed to facilitate PEG chain threading through the CD rings. Hydrogels then formed via Michael addition between thiol and maleimide upon adding PEG-Mal. Hydrogels with varying PEG-SH to PEG-CD ratios were prepared, using hydrogels without PEG-CD as controls. PEG-SH

can thread through the CD rings due to the smaller diameter of the thiol group (~1.3 Å) compared to the inner cavities of α-CD (~4.5 Å) and β-CD (~6.1 Å) (Supplementary Fig. 8a). More than 95% of PEG-CD integrated into the network (Supplementary Fig. 8b, c), with nearly 90% of CD rings filled by PEG chains (Supplementary Fig. 8d, e), indicating a high density of sliding structures. All hydrogels showed similar swelling ratios, water contents, porous microstructures and mesh sizes (Supplementary Figs. 9 and 10).

Mechanical properties of these hydrogels were assessed by compressive tests. The modulus slightly increased in the presence of PEG-CD (Supplementary Fig. 11a–c), while toughness notably improved, likely due to enhanced energy dissipation (Supplementary Fig. 11d). Notably, the presence of PEG-CD enhanced the fracture strain of hydrogels, likely due to that the chain walkers moving along the PEG chains at the break can also dissipate energy, thereby preventing crack propagation at the break point and increasing the break strain. Moreover, the chain walkers present on distinct tetra-PEG molecules can act as mobile crosslinks for the end free polymers that do not form covalent crosslinks, thus altering network connectivity akin to those observed in typical slide-ring hydrogels. At high strains, the mechanical stresses in hydrogels containing PEG-CD are greater than those in hydrogels without PEG-CD. This difference can be attributed to that parts of the chain walkers may act as fixed crosslinking points after

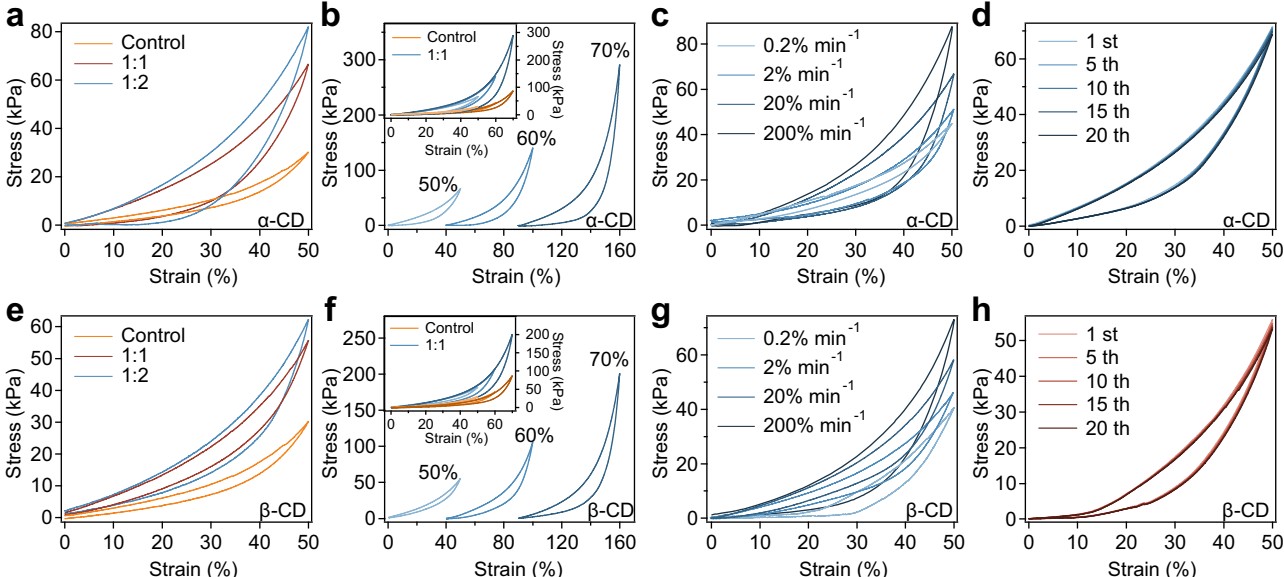

**Fig. 3 | Energy dissipation and fast recovery of hydrogels based on the molecular friction of chain walkers. a** Typical compression-relaxation curves of hydrogels at different PEG-SH:PEG-α-CD ratios (1:1 and 1:2) at a strain of 50%. The hydrogel prepared without chain walkers was used as the control. **b** Compression-relaxation curves of hydrogels at various strains (PEG-SH:PEG-α-CD = 1:1). **c** Compress-relaxation curves of hydrogels at different deformation rates (0.2% min⁻¹, 2% min⁻¹, 20% min⁻¹ and 200% min⁻¹) at a PEG-SH:PEG-α-CD ratio of 1:1 (strain ~50%). **d** Consecutive compression-relaxation cycles of hydrogels containing PEG-α-CD without any waiting time for 20 cycles at the deformation rate of 20% min⁻¹ (PEG-SH:PEG-α-CD = 1:1). **e** Typical compression-relaxation curves of hydrogels at

different PEG-SH:PEG-β-CD ratios (1:1 and 1:2) at a strain of 50%. The hydrogel prepared without chain walkers was used as the control. **f,** Compression-relaxation curves of hydrogels at various strains (PEG-SH:PEG-β-CD = 1:1). **g** Compression-relaxation curves of hydrogels at different deformation rates (0.2% min⁻¹, 2% min⁻¹, 20% min⁻¹ and 200% min⁻¹) at a PEG-SH:PEG-β-CD ratio of 1:1 (strain ~50%). **h** Consecutive compression-relaxation cycles of hydrogels containing PEG-β-CD without any waiting time for 20 cycles at the deformation rate of 20% min⁻¹ (PEG-SH:PEG-β-CD = 1:1). Each experiment was repeated 3 times independently with similar results.

sliding to stable states during deformations. Compression-relaxation cycles revealed clear hysteresis in hydrogels with PEG-CD, in contrast to minimal hysteresis in control hydrogels (Fig. 3a). Hydrogels with PEG-α-CD at PEG-SH:PEG-α-CD ratios of 1:1 and 1:2 showed energy dissipation ~4.2 and ~6.2 times higher than controls at 50% strain (Supplementary Fig. 12a), with relative energy dissipation reaching 42.0% and 49.2% (Supplementary Fig. 12c). Hydrogels with PEG-β-CD at similar ratios exhibited energy dissipation ~1.6 and ~3.3 times the controls (Fig. 3e and Supplementary Fig. 12b), with relative dissipations of 28.2% and 37.0% (Supplementary Fig. 12d). Increased strains enhanced hysteresis due to longer sliding distances (Fig. 3b, f), with energy dissipation for hydrogels containing PEG-α-CD and PEG-β-CD rising by 3.1 and 3.5 times, respectively, from 50% to 70% strain (Supplementary Fig. 13a, b). At 70% strain, relative energy dissipation reached 49.2% and 40.0%, significantly exceeding the 26.7% in control hydrogels (Supplementary Fig. 13c, d). Moreover, hydrogels containing CD monomers that were not linked via the PEG segment were also prepared for comparison. As shown in Supplementary Fig. 14, the mechanical strength and energy dissipation of these hydrogels were almost the same as those of control hydrogels, suggesting that introducing CD monomers cannot affect energy dissipation of hydrogels.

Furthermore, we observed that the energy dissipation in hydrogels correlates positively with the deformation rate, consistent with the trends seen in molecular frictional forces at varying pulling speeds (Fig. 3c, g). Specifically, hydrogels embedded with PEG-α-CD exhibited a 400% increase in energy dissipation when the deformation rate was altered from 0.2% to 200% (Supplementary Fig. 15a). Hydrogels containing PEG-β-CD showed a slightly lower increase of 340% (Supplementary Fig. 15b). In contrast, the energy dissipations of the control hydrogels remained almost the same under different deformation rates, suggesting negligible rate-dependent properties (Supplementary Fig. 16). This indicates that higher molecular frictional forces

contribute to larger energy dissipation at a macroscopic scale. We also discovered that the energy dissipation is affected by the length of the PEG-CD linkers in the chain walkers. Extending the PEG linkers from 2 kDa to 5 kDa significantly lowered the energy dissipation in hydrogels, bringing it close to the levels in hydrogels without chain walkers (Supplementary Fig. 17). This implies that the leg length of chain walkers needs to be shorter than the network strands for effective friction. If the legs are excessively long, the hydrogel network's deformation fails to sufficiently stretch the chain walker legs, thereby reducing friction with the hydrogel network.

We also examined the recovery of energy dissipation through cyclic compression-relaxation tests of the hydrogels (Fig. 3d, h). Stress-strain curves of hydrogels at a PEG-SH:PEG-CD ratio of 1:1 were almost identical across successive cycles. After 20 cycles, the maximum stress and dissipated energy retention exceeded 94% for both hydrogel types, showcasing rapid recovery of energy dissipation and mechanical strength within seconds (Supplementary Fig. 18). In contrast, the energy dissipations of the control groups remained low and decreased slightly over 20 cycles of compression-relaxation (Supplementary Fig. 19). Moreover, the energy dissipation of damping hydrogels decreased to ~89% after 20 cycles of stress-relaxations at the higher loading rate (200% min⁻¹), which can be attributed to that the chain walkers may not fully restore their positions at high deformation rates (Supplementary Fig. 20). These findings collectively underscore the crucial design of our hydrogel network in enhancing both the energy dissipation and quick recovery capabilities of hydrogels.

## Damping hydrogel protects cells under dynamic loading
The energy dissipation of damping hydrogels, driven by molecular friction, offers protection to encapsulated cells, emulating the shielding properties of natural soft tissues[65]. To demonstrate this, human mesenchymal stem cells (hMSCs) were embedded in hydrogels

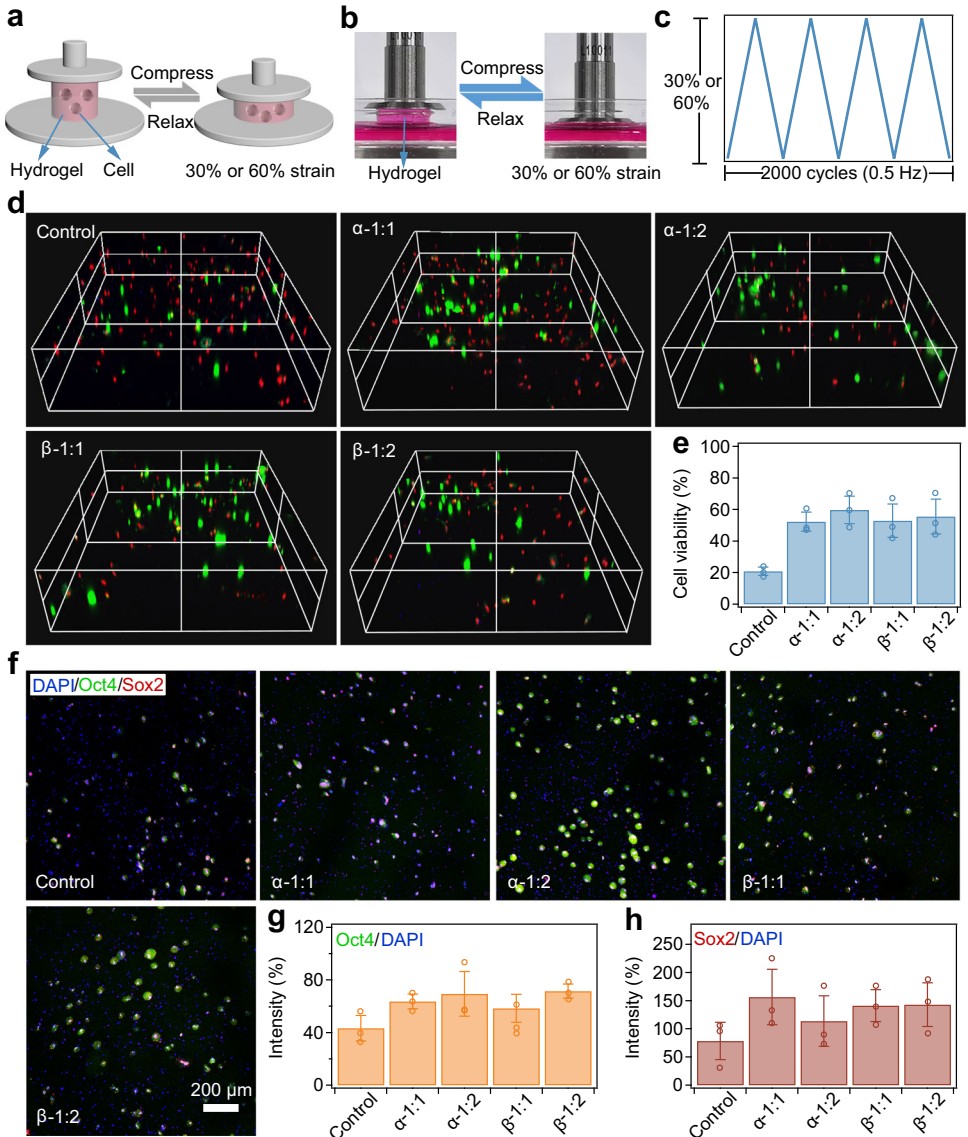

**Fig. 4 | Damping hydrogels protect hMSCs from damage and loss of stemness under dynamic loading. a** Illustration of the protection of hMSCs in hydrogels under cyclic compression-relaxation. **b, c** Optical images (**b**) and strain signals (**c**) for cyclic compression-relaxation of hydrogels with hMSCs encapsulated inside. Conditions: frequency ~0.5 Hz, cycle number ~2000, strain ~30% or 60%. **d** 3D reconstructions of live/dead cell staining in different hydrogels using laser confocal fluorescence microscopy (LCFM) after 2000 cycles of compression-relaxation (strain: 60%). Cells were stained using calcein-AM (green) and propidium iodide (PI) (red). The size of the scanning space was 1272 μm × 1272 μm × 300 μm. Each experiment was repeated 3 times independently with similar results. **e** Cell viabilities of hMSCs in different hydrogels after cyclic compression-relaxation. Values represent the mean and standard deviation (sample size $n = 3$). **f** Stemness of hMSCs in hydrogels ($C_{PEG\text{-}Mal} = C_{PEG\text{-}SH} = 70$ mg mL$^{-1}$) after cyclic compression-relaxation (strain: 60%). A space of 1272 μm × 1272 μm × 300 μm was scanned, and the projected image in the Z-axis direction is shown. Specific markers for stemness maintenance, Oct4 (green) and Sox2 (red), were stained. Cell nuclei are indicated by DAPI (blue). Each experiment was repeated 3 times independently with similar results. **g, h** Normalized intensities of Oct4 (**g**) and Sox2 (**h**) for hMSCs in hydrogels after cyclic compression-relaxation (strain: 60%). The intensity of DAPI was set as 100%. Values represent the mean and standard deviation (sample size $n = 3$).

containing PEG-CD during their preparation. These cylindrical hydrogels were then subjected to cyclic compression-relaxation at 0.5 Hz, with moderate (30%) and excessive (60%) strains (Fig. 4a–c). Three-dimensional (3D) live/dead staining (Fig. 4d, e) revealed significantly higher live cell densities in hydrogels with PEG-CD than those without PEG-CD after 2000 cycles at ~60% strain. Cell viability in hydrogels with PEG-CD generally exceeded 50%, at least double that of hydrogels without PEG-CD (Fig. 4e and Supplementary Fig. 21). Notably, hydrogels containing PEG-α-CD showed slightly higher cell viability than those with PEG-β-CD, likely due to greater energy dissipation. The presence of dead cells in hydrogels that did not undergo cyclic compression-relaxation can be attributed to internal stress from the hydrogel network during swelling, rather than the

cytotoxicity of the hydrogels, which was found to be negligible (Supplementary Fig. 22).

Additionally, the cell shielding effect correlated directly with energy dissipation amplitudes. Cell viability enhancements post-cyclic compression-relaxation decreased with lower deformation strain due to the reduced energy dissipation (Supplementary Fig. 23). Hydrogels not subjected to compression maintained consistent cell viability, suggesting that cell protection is not merely due to CD addition (Supplementary Fig. 24). Hydrogels with lower solid contents also exhibited cell shielding, albeit with slightly diminished viability enhancements, possibly due to a lower crosslinking density increasing inherent viability (Supplementary Figs. 25–27). Notably, hydrogels at lower solid contents also exhibited stable geometry and microporous

structures, despite the much weaker mechanical strength (Supplementary Fig. 28).

Furthermore, the permanent deformations of different hydrogels were lower than 5% after 2000 cycles of compression-relaxation (Supplementary Fig. 29). The loss moduli ($G''$) were significantly smaller than the storage moduli ($G'$), and no notable stress relaxation was observed for both damping and control hydrogels. This suggests that the mechanical response of the hydrogels predominantly originates from the covalently crosslinked network[66–69], as shown in Supplementary Fig. 30. This contrasts sharply with the behavior of physically crosslinked networks, which exhibit noticeable stress relaxation due to the rupture of physical crosslinks. These findings demonstrate that the enhanced energy dissipation in hydrogels, attributed to molecular frictions, effectively guards encapsulated stem cells against excessive mechanical damage under loading. The efficient and rapid recovery of damping hydrogels during multiple stress-relaxation cycles release mechanical load efficiently with each cycle, further ensuring effective protection of cells under continuous compression-relaxation cycles.

Considering that the mechanical history significantly influences stem cell stemness[70–72], we explored whether our damping hydrogels could preserve the stemness of hMSCs after dynamic loading. Our findings revealed that the stemness of hMSCs was effectively maintained post-compression, as evidenced by immunostaining for Oct4 and Sox2. These transcription factors are crucial for stem cell self-renewal and pluripotency[73,74]. Notably, the expression levels of Oct4 and Sox2 were markedly higher in hMSCs encapsulated within PEG-CD hydrogels compared to those in hydrogels without PEG-CD (Fig. 4f–h and Supplementary Fig. 31). These outcomes indicate that our damping hydrogels provide a conducive mechanical environment that supports the maintenance of stem cell stemness, highlighting their potential utility in applications where mechanical regulation of stem cell behavior is critical.

## Discussion

In this work, we report a hydrogel design, integrating molecular friction to achieve both rapid mechanical energy damping and swift recovery, a feat not attainable with conventional methods. Traditional hydrogels often rely on viscoelastic properties, using the rupture of sacrificial bonds or networks to dissipate energy[11,12]. However, this approach falls short under dynamic loading due to the lack of a mechanism for fast recovery, leading to significantly diminished damping capacity over multiple cycles[16,19,24,26]. Our friction-based mechanism overcomes this limitation. By introducing molecular friction into the hydrogel matrix, we enable both efficient energy dissipation and rapid recovery. This ensures that the hydrogel maintains its damping capabilities even after repeated loading, a critical advancement for applications requiring sustained performance under dynamic conditions.

Typical hydrogels face a trade-off between strength and toughness. The strength is directly proportional to the concentration of the polymer (c), while toughness inversely relates to the square root of this concentration ($c^{-1/2}$)[75,76]. In our hydrogel design, we decouple energy dissipation from strength, presenting a mechanism that enhances toughness independent of polymer concentration and crosslinking types. This innovative approach allows us to design hydrogels that are both strong and tough, overcoming a longstanding challenge in hydrogel engineering. Typically, damping effects in hydrogels are attributed to viscoelastic dissipation mechanisms, which incorporate sacrificial physical bonds formed through various interactions such as hydrophobic interactions, ionic interactions, hydrogen bonding, coordination interactions, and host-guest interactions[77–82]. The reformation of these physical bonds is influenced by the rebinding kinetics and the entropic recoiling of polymer strands, processes that are generally slow. This delay hampers the rapid restoration of energy

dissipation capacity, especially under multiple load/unload cycles. In contrast, our design leverages the friction of chain walkers within the polymer network, bypassing the need for the rupture and reformation of physical bonds and avoiding alterations to the hydrogel network structure. We have shown that the fast repositioning of chain walkers via diffusion quickly restores the hydrogel's energy dissipation capabilities within seconds, ensuring sustained damping effects during repeated load/unload cycles.

It is also worth mentioning that achieving molecular friction and fast recovery of CD positions is not straightforward and requires a carefully designed hydrogel network. A key factor is the mismatch between the lengths of the chain walker legs and the track, which creates friction during stretching. Additionally, the predefined range of walker movement, set by covalent crosslinks, is critical for rapid recovery of CD positions, which facilitates the fast restoration of the damping capacity in our design. These features distinguish our hydrogels from traditional slide-ring hydrogels, where CD recovery is not crucial for achieving their exceptional mechanical properties.

The mechanical properties of our hydrogels, particularly their ability to rapidly dissipate mechanical energy and recover, open up a wide range of potential biomedical applications. For example, the enhanced energy dissipation and rapid recovery characteristics make these hydrogels ideal for impact-resistant medical devices or protective gear, where they can absorb mechanical shocks while maintaining structural integrity. Furthermore, their biocompatibility and tunable properties open avenues in tissue engineering, particularly in developing scaffolds that mimic the dynamic mechanical environment of natural tissues. They could also be instrumental in drug delivery systems where controlled release is essential, particularly in dynamic physiological conditions. In addition, the properties of these hydrogels have potential applications in creating soft robotics components, which require materials that can endure and quickly recover from high strains. We also envision that the molecular friction mechanism could inspire more designs in prosthetics and orthopedic implants, offering better durability and comfort for users.

In summary, the integration of molecular friction into hydrogel design not only solves existing limitations of hydrogels in dynamic environments but also expands the potential of these versatile materials in various biomedical applications.

## Methods
### Materials
Maleimide-terminated 4-armed PEG (4-armed PEG-Mal, Mw: 20 kDa), thiol-terminated 4-armed PEG (4-armed PEG-SH, Mw: 20 kDa), PEG with one end terminated with maleimide and the other end terminated with methyl (mPEG-SH, Mw: 10 kDa), and succinimidyl-terminated PEG (NHS-PEG-NHS, Mw: 2 and 5 kDa) were purchased from SinoPEG, Inc. Amino-modified cyclodextrins ($\alpha$-CD-NH$_2$ or $\beta$-CD-NH$_2$) were purchased from TCI (Shanghai). The cell culture medium $\alpha$-MEM, fetal bovine serum, streptomycin/penicillin, and L-glutamine were purchased from Wisent, Nanjing, China. (3-Aminopropyl) triethoxysilane (APTES), trimethoxysilylpropanethiol (MPTMS) and succinimidyl 4-(N-maleimidomethyl) cyclohexane-1-carboxylate (SMCC) were purchased from Thermo Scientific. Other reagents related to the cell culture included Tween-20 (cat: CT371, U-CyTech, USA), Triton X-100 (cat: 0694, Amresco, USA), primary antibodies, including anti-Sox2 antibody (cat: D9B8N, CST, USA), anti-Oct4 antibody (cat: D7O5Z, CST, USA), 4',6-diamidino-2-phenylindole (DAPI, cat: MBD0015, Sigma–Aldrich, China), Alexa Fluor 647-labelled goat anti-rabbit IgG (H + L, cat: A0468, Beyotime, China) and FITC-labelled goat anti-mouse IgG (H + L, cat: E031210, Earthox, USA). Unless otherwise stated, all the other reagents were purchased from Sinopharm Chemical Reagent Co., Ltd. (Beijing, China). Cell line human mesenchymal stem cells (hMSCs, cat: PCS-500-012) were purchased from Beijing Zhongyuan Ltd. (Beijing, China), which are authenticated by STR profiling.

## Preparation of the PEG-coated cantilevers and CD-coated substrates

Standard silicon nitride ($Si_3N_4$) cantilevers were obtained from Bruker (type: MLCT). The cantilevers were immersed in chromic acid for 20 min at 95 °C and washed with Milli-Q water. Then, the cantilevers were immersed in 1% (v/v) APTES methylbenzene solutions for 1 h to introduce amino groups to the surface, followed by rinsing with methylbenzene and ethanol. After drying under nitrogen, the cantilevers were immersed in DMSO containing 1 mg mL$^{-1}$ SMCC for 1 h. Then, the cantilever was dried using nitrogen and immersed in DMSO containing 1 mg mL$^{-1}$ thiol-terminated PEG (mPEG-SH, 10 kDa) for another 1.5 h. Finally, the PEG-coated cantilevers were washed with DMSO and Milli-Q water before drying with nitrogen.

For the CD coated substrates, the glass substrates were treated with chromic acid for 12 h. Then, the substrates were immersed in a methylbenzene solution containing MPTMS (0.5 mg mL$^{-1}$) for 1 h to introduce the thiol group. Subsequently, the substrates were rinsed with methylbenzene and ethanol, dried under nitrogen and immersed in DMSO containing SMCC (1 mg mL$^{-1}$) for 1 h. Finally, the substrates were rinsed with DMSO and ethanol, dried under nitrogen and immersed in DMSO solutions containing amino-modified α-CD or β-CD (0.1 mM) for 1 h. After washing with DMSO and ethanol again, the α-CD- or β-CD-coated substrates were dried with nitrogen. All the cantilevers and substrates were used immediately as soon as the preparation was completed.

## Single-molecule force spectroscopy experiments based on AFM

Single-molecule force spectroscopy (SMFS) experiments were carried out on a commercial AFM (JPK Nanowazrid II) in PBS solutions (10 mM, pH = 7.4) at room temperature (~25 °C). The PEG-linked cantilevers (spring constant of ~0.05 N m$^{-1}$) were used in all experiments. The spring constant was calibrated using the equipartition theorem for each experiment. During each SMFS experiment, the cantilever was brought into contact with the surface at a speed of 3.2 μm s$^{-1}$ to a contact force of ~500 pN for 200 ms to allow PEG to thread through the CD ring. Then, the cantilever was pulled back at the same speed to obtain the force-extension curves. All force curves were collected by commercial software from JPK and analysed offline using a custom-written protocol in Igor 6.12 (Wavemetrics, Inc.). The dynamic force spectroscopy experiments were also performed with different pulling speeds (200, 400, 800, 1600 and 3200 nm s$^{-1}$).

## Molecular dynamic (MD) simulation of frictions

The force fields of α-CD, β-CD and PEG (Mw: 1529.8, N = 34) were generated using the general AMBER force field (GAFF)[83] and Antechamber[84]. The molecular configurations were obtained from the PDB bank (PEG: http://www.rcsb.org/ligand/15P; α-CD: http://www.rcsb.org/ligand/ACX; β-CD: http://www.rcsb.org/ligand/BCD). The whole system was in a TIP3P water box with temperature and pressure controlled by a Langevin thermostat and Berendsen barostat. The initial state was generated at 500 k, and a carbon atom of the CD ring was fixed. The simulation of pulling PEG was performed from the initial state at 300 K. The work done by the pulling force was determined as $U = k(|r_0 - r_1| - r)^2$, in which $r_0$ is the reference point and $r_1$ is the head of PEG. $r$ is defined as $r = |r_0 - r_1|_{t=0} \times (1 - t/t_0)$, where $t_0$ corresponds to the time used for pulling out of PEG through CD. The pulling forces were calculated as the slope for the curves of work vs pulling distance, and the force-distance curves were the means of multiple curves. The number of hydrogen bonds during the pulling of PEG was also analysed.

## Preparation of hydrogel

For the preparation of hydrogels without PEG-CD, 4-armed PEG-Mal and 4-armed PEG-SH were dissolved in ddH$_2$O to a concentration of 7 mM. Then, the two kinds of solutions were quickly mixed at a volume ratio of 1:1. Transparent hydrogels were formed in seconds after mixing. Then, the gels were dialyzed in ddH$_2$O for 24 h to allow swelling equilibrium. For the preparation of hydrogels containing CD monomers, CD was dissolved in 4-armed PEG-SH solutions (7 mM) in varying proportions (PEG-SH: CD = 1:2 or 1:4), and the solutions were stored at room temperature for 4 h to allow the PEG chain to thread through the ring. Then, the resulting solution was mixed with 4-armed PEG-Mal solution (7 mM) at a volume ratio of 1:1. Transparent hydrogels formed after mixing and dialyzed in ddH$_2$O for 24 h to allow swelling equilibrium. For the preparation of hydrogels containing PEG-CD, PEG-CD was dissolved in 4-armed PEG-SH solutions (7 mM) in varying proportions (PEG-SH:PEG-CD = 1:1 or 1:2), and the solutions were stored at room temperature for 4 h to allow the PEG chain to thread through the ring. Then, the resulting solution was mixed with 4-armed PEG-Mal solution (7 mM) at a volume ratio of 1:1. Transparent hydrogels formed after mixing and dialyzed in ddH$_2$O for 24 h to allow swelling equilibrium. For the preparation of hydrogels with lower solid contents, the concentrations of 4-armed PEG-Mal and 4-armed PEG-SH in solutions before mixing were 2 mM and the hydrogels were prepared similarly as described above.

## Compressive test

The compressive stress–strain measurements were performed using a tensile-compressive tester (Instron-5944 with a 2 kN sensor) in air. In compression-crack and compression-relaxation tests, the rate of compression was kept constant at 20% min$^{-1}$ with respect to the original height of the hydrogel, roughly in a range of 1.6–2.0 mm min$^{-1}$. The stress ($\sigma$) was calculated as the compression force divided by the cross-sectional area, which was monitored by a side view CCD camera during the compression process. The toughness ($E_f$) was calculated by the integration of the area below the compression force-distance curves until the fracture point. The equation used in the calculation was as follows: $E_f = \int_{x_0}^{x_f} \sigma(x)dx$, in which $x_0$ and $x_f$ correspond to the starting distance and the fracture distance of the compression, respectively. The Young's moduli were the approximate linear fitting values of the stress–strain curves in the strain range of ~20%. The dissipated energy ($E$) was calculated as the area enclosed by each compression-relaxation cycle. The relative energy dissipation was determined as $\varphi = \frac{E}{E_0} \times 100\%$, in which $E$ corresponds to the dissipated energy and $E_O$ is calculated as follows: $E_0 = \int_{x_0}^{x_e} \sigma(x)dx$, in which $x_0$ and $x_e$ correspond to the starting strain and the maximum strain of the compression, and $\sigma(x)$ corresponds to the compression curve.

## Shielding encapsulated cells in hydrogels under dynamic loading

For the 3D cell culture in hydrogels, hMSCs were washed from the cell culture plates and mixed into precursor solutions at a density of $5 \times 10^4$ mL$^{-1}$. All the solutions were filtered with PVDF (0.22 μm, Millipore). Then, the hydrogels were prepared as previously described. The hydrogels were immersed in cell culture medium (α-minimum essential medium containing 10% fetal bovine serum, 1% streptomycin/penicillin, and 1% L-glutamine), and the cells were cultured for 24 h. Then, the hydrogels were forced to undergo cyclic compression (0.5 Hz) at different strains in cell culture medium. After 2000 cycles of compression, the cell viability inside the hydrogels was investigated using a live/dead viability/cytotoxicity kit (Calcein-AM/PI Double Staining Kit). Following a triple wash with PBS solution, a mixture of calcein-AM (diluted 1:500) and propidium iodide (PI) (diluted 1:500) dye solution was added to the hydrogels. Subsequently, the hydrogel was incubated at 37 °C for 30 min before undergoing three additional washes with PBS. Finally, images were obtained using a laser confocal fluorescence microscope (Olympus FV3000, Japan). Typically, a space of 1272 μm × 1272 μm × 300 μm was scanned layer by layer at a step size of 2 μm. The images were projected onto the z-axis, and the cell

viability was determined by analysing the images using ImageJ (Version 1.8.0).

## Immunostaining staining and analysis

For immunofluorescence, hydrogels with hMSCs encapsulated inside were fixed in 2% (vol/vol) paraformaldehyde for 30 min and then treated with 0.25% Triton X-100 for 15 min. After blocking in 1% (wt/vol) BSA for 1 h to minimize nonspecific binding, anti-Oct4 (diluted 1:200) and anti-Sox2 (diluted 1:200) diluted with PBST (0.5 wt% Tween-20 in PBS) containing 1% (wt/vol) BSA were added to the fixed cells and incubated overnight at 4 °C. Then, the primary antibody solution was decanted, and the dish was immediately washed with PBS 3 × 5 min. After being rinsed with PBST (0.5 wt% Tween-20 in PBS) 3 times, the secondary antibody (FITC-labelled goat anti-mouse, Alexa Fluor 647-labelled goat anti-rabbit) (diluted 1:200) in PBST containing 1% (wt/vol) BSA was added to the dish, and the cells were incubated for 60 min at room temperature in the dark. Then, the secondary antibody was decanted, and the cells were washed with PBST 3 × 5 min. All images were obtained using a laser confocal fluorescence microscope (Olympus FV3000, Japan). Quantitative image analyses were performed using ImageJ (v1.8.0).

## Reporting summary

Further information on research design is available in the Nature Portfolio Reporting Summary linked to this article.

## Data availability

All data are available in the main text or the Supplementary Information. Data is available from the authors on request. Source data are provided with this paper.

## Code availability

The custom-written protocol in Igor 6.12 is available from the authors on request.

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

## Acknowledgements

This research is supported mainly by the National Natural Science Foundation of China (Grant Nos. T2322010 (B.X.), 11934008 (W.W.), T2225016 (Y.C.), and T2222019 (H.L.)), the National Key R&D Program of China (Grant No. 2020YFA0908100 (Y.C.) and 2023YFC3605802 (Y.L.)), the Natural Science Foundation of Jiangsu Province (Grant No. BK20220120 (B.X.)), and Fundamental Research Funds for the Central Universities (Grant No. 020514380274 (B.X.)).

## Author contributions

W.W., Y.C., and B.X. conceived the idea and designed the study. Z.X. performed the experiments. J.L. and W.L. performed the MD simulations. Z.X. and B.X. analyzed the results. D.L., Y.L., H.L. and B.C. help design and discussed the experiments. B.X., Y.C. and Z.X. wrote and refined the paper. W.W., Y.C. and B.X. supervised the project. All the authors discussed the results.

## Competing interests

The authors declare no competing interests.
