## [Peer Review File · Nature Communications]

Rapidly damping hydrogels engineered through molecular frictionREVIEWER COMMENTS

Reviewer #1 (Remarks to the Author):

The authors combine two previously reported designs of hydrogels: tetra-arm PEG hydrogels and slide-ring hydrogels. They claim that this results in a ‘damping behavior’, achieving both compression-based hysteresis and good cyclical behavior. Additionally, the authors present cell-based viability studies during compressive deformations. While the improvements in cell viability is an interesting highlight of the work, it still requires further explanation and control experiments before it can be considered for published in Nature Communications.

Major comments:

1. The authors claim that the ‘chain walkers’ can be rapidly restored to their initial state via diffusion. Can the authors show whether diffusional forces could overcome the friction-based forces they have calculated from SMFS?
2. As mentioned in line 107, “the conventional slide-ring hydrogels demonstrated minimal hysteresis and energy dissipation”. The minimal hysteresis is actually a sign of rapid recovery. For example, almost 100% rapid recovery (after 100th consecutive tensile cycles) and self-reinforcement is realized in previously reported slide-ring hydrogels, for which you can refer: Liu et al., Science 372, 1078–1081 (2021). Consequently, figure 1a (i) and (iii) showing the “un-recovery” of conventional slide-ring hydrogels is not true. The authors must properly distinguish the cyclability/recovery ability of their design and the conventional slide-ring hydrogels in the main text as well as in Figure 1?
3. The authors must explain why the strain at break is improved by the ‘chain walkers’, compared to the control hydrogel (Supplementary Figure 11a)? Presumably only the modulus would be increased (by the friction of chains sliding through cyclodextrin), while the strain would still be limited by the covalent PEG crosslinks.
4. There should be further experimental comparisons performed with the control gels. How does the successive cycling and rate-dependent behavior of the control gels compare?
5. It is unclear whether the improved cell viability is only due to a damping effect. It is well documented that viscoelasticity in hydrogels can improve cell viability via stress relaxation. The authors should perform rheological experiments to determine the difference in stress relaxation and storage/loss modulus viscoelastic properties between the damping gel and the control PEG gels.
6. Is the mechanical cycling performance rate dependent? At higher rates of compression/relaxation, one would expect that the ‘chain walkers’ may not have enough time to

diffuse to their low energy state between cycles?

7. Is there any permanent deformation experienced by either of the gels during this 2000 cycling experiment? If the control hydrogel is undergoing permanent deformation over cycles, this may explain the poor cell viability.

Minor comments:

1. In Figure 1, should the direction of the deformation and relaxation arrows be swapped?
2. The authors need to include an explanation for their energy dissipation calculation.
3. While Figure 1d seems understandable to interpret the CD-PEG-CD as a “chain walker”, the reviewer had a hard time understanding the network structure changes during deformation with Figure 1f. It would be helpful to indicate the motion direction for either the CD ring or the PEG chain. Can the authors please improve this schematic?

Reviewer #2 (Remarks to the Author):

In the manuscript by Zhengyu Xu, Jiajun Lu, et al., the authors introduce a molecular-level damping mechanism into hydrogels by embedding slide-ring-based intra- and inter-chain linkers. This mechanism can achieve relatively substantial energy dissipation, as well as rapid recovery, which is attributed to the fast dynamics of bridged slide-rings on the polymer backbone. The authors further demonstrate that the hydrogels they designed can effectively shield encapsulated cells from mechanical trauma under cyclic compression, holding potential for use as biomaterials in dynamic loading circumstances. Overall, the design of this work is interesting, but the interpretation of the experimental results could be improved. I would reconsider it for publication in Nature Communications after major revision. Here are several points that might be helpful:

1. In lines 52-54, there is a very important concept that the authors need to clarify. The sacrificial bonds or interactions themselves do not store too much elastic energy. Instead, it is the polymer chains that store the majority of the elastic energy, which gets released upon the scission of the sacrificial bonds.
2. In lines 54-56, the statement “However, to enhance energy dissipation, the energy of these sacrificial bonds/networks needs to be substantial, which can compromise the dynamic properties of the hydrogel” is misleading. It is not the substantial energy of the sacrificial bonds that compromises the dynamic properties of the hydrogels; rather, it is the slow reformation and the inability to find the original pair that compromise the fast recovery of the gels.
3. In lines 82-84, what limits the mobility of the CD rings in slide-ring gels under stress? The slide-ring gels developed by Ito and coworkers are very resilient and can fully recover their original

shapes after large deformation. The lack of an energy dissipation mechanism in the gels is due to their high elasticity under typical strain rates, not because they cannot recover. The picture the authors describe is different from my understanding of slide-ring gels. The authors should provide references for this statement.

4. In lines 103-106, the free energy of the typical slide-ring gels at small deformation comes from two parts: the entropy of the rings and the entropy of the chains. I am skeptical about the schematic potential profile shown in figure 1b. There should be no significant free energy difference between states i and iii. Here are two main reasons: Firstly, without external load, the chains are in their ideal state. The chain between crosslinks is always dominated by thermal fluctuation (kT). Since the number of chains is the same in states i and iii, they should have similar free energy. States i and iii might have slightly different free energy in terms of the rings, but that is trivial. Secondly, if figure 1a is accurate, slide-ring gels would have a higher loading Young's modulus than their unloading Young's modulus since modulus indicates the free energy density within the networks. This is not observed in the system developed by Ito and coworkers. I can imagine the rings might not be able to return to their original position after deformation, but it does not mean the gel cannot recover its original shape. The number of chains remains unchanged.

5. In Figure 1e, I do not believe state iii is unstable. There should be some activation barrier to overcome to return to state i. The barrier is just smaller than that in the case of figure 1a, and it occurs very fast under thermal fluctuation.

6. Based on the gel synthesis procedure described by the authors, there should be both intra- and inter-chain slide-ring linkers. Figure 1d shows the former, while Figure 1e shows the latter. Although they might not be significantly different, it would be beneficial if the authors could clarify that they are not the same.

7. In the section on SMFS, is there a difference in the success rate of threading a PEG chain through alpha-CD versus beta-CD? This is a minor point, but it piques my curiosity.

8. In figure 3a, the unloading curve of the 1:2 ratio is lower than that of the control, suggesting that the elastically active structure of the 1:2 gel is likely less than that of the control. There could be more defects in the 1:2 gel than in the control, so some part of the hysteresis of the 1:2 gel probably has contributions from dangling defects. It is probably beyond the scope of this paper, but it would be truly helpful if the authors could confirm that the elastically active structures of the different gels are the same, making the dissipation mechanism they proposed more convincing.

9. In figures 3d and h, the authors should indicate the loading rate.

10. It is not very clear to me how the fast recovery of the gel protects cells under dynamic loading. In reference 59, energy dissipation offers protection to encapsulated cells, but the role of fast dynamics is a bit vague. Can the authors explain this in more detail? Can they clarify why their design is better than that in reference 59?

11. In the section on the preparation of hydrogel, the authors state that the gels were prepared at 7 mM or 2 mM. I wonder which concentration was used for figure 3? The molecular weight of the tetra PEG is 20 kDa, and 2 mM (40 mg/mL) is just at the overlap concentration. How well is the gel formed under this condition?

12. I wonder if the authors can make a PEG gel using the same procedure but use only CD molecules instead of PEG-CD? A network with CD rings on the PEG backbone but without the bridging PEG chain would be a better control compared to the current one.

Reviewer #3 (Remarks to the Author):

The manuscript presents an innovative approach to designing hydrogels for swift mechanical energy dissipation with a friction-based damping mechanism. By introducing molecular friction into the hydrogel matrix, the efficient energy dissipation and rapid recovery were achieved in a same hydrogel. The authors highlighted the potential applications of such hydrogels in impact protection and shock absorption such as shielding encapsulated cells during cyclic compression-relaxation. The authors explored the effective energy dissipation based on the slide-ring friction, which is really new and often overlooked in conventional hydrogels. They also emphasize the design principle of hydrogel network that facilitates the motion of a 'chain walker', leading to the efficient molecular friction.

Overall, the research is well-designed, and the experiments and simulations are carefully executed, supporting the conclusions drawn. It can provide significant insights into designing the mechanical energy dissipation of hydrogels, potentially impacting various practical applications in biomechanics and related fields. I recommend the publication of this manuscript after a few minor revisions.

1. As depicted in Figure 1, it seems to me that the CD ring can slide along the PEG chain during the initial stretching. Considering that the free energy level at the initial state is higher than the final state, could the first round of deformation also induce ring sliding, and could this process dissipate energy to some extent? It might be helpful for the authors to provide clarification on this point.

2. In the MD simulation of sliding friction and molecular interactions, the frictional forces were observed to be much higher than those measured in SMFS. The authors attributed this difference to the significantly greater pulling speeds used in simulations. It would be beneficial for the authors to include a detailed discussion on this point, supported by relevant references.

3. Would the introduction of CD-PEG-CD affect the initial bulk properties of the hydrogel, such as pore size? Since the authors have stated that the hydrogels exhibited similar porous microstructures, it would be helpful to provide a detailed distribution of the mesh size.

4. The mechanical stresses observed in hydrogels containing PEG-CD at high strains, such as 50% or 60%, are greater than those in hydrogels without PEG-CD. What could be the reason for this difference? Does it suggest that the CD dimer's chain walker may also function as a crosslinking agent after being stretched under significant deformations?

5. In the demonstration of protecting hMSCs from damage and loss of stemness under dynamic loading, the labels on the figures should specify α -CD or β -CD for clarity, rather than just α or β . Additionally, it seems that there were dead cells in hydrogels that did not undergo cyclic compression-relaxation. What might be the reason for this observation? Could it be due to internal stress from the hydrogel network during swelling, or could it be related to the cytotoxicity of the hydrogel? To address this question, it is recommended to include a cytotoxicity experiment.

6. In the images of Figure 4f, there are some instances of mismatched cell staining and noisy

points. It would be preferable to replace these images with versions featuring clear backgrounds.

Point-by-point response to the reviewers' comments

Reviewer #1 (Remarks to the Author):

The authors combine two previously reported designs of hydrogels: tetra-arm PEG hydrogels and slide-ring hydrogels. They claim that this results in a ‘damping behavior’, achieving both compression-based hysteresis and good cyclical behavior. Additionally, the authors present cell-based viability studies during compressive deformations. While the improvements in cell viability is an interesting highlight of the work, it still requires further explanation and control experiments before it can be considered for published in Nature Communications.

General Response: We thank the reviewer for the insightful comments. In order to address her/his concerns in full, we have conducted a substantial revision with new data, analyses and detailed methodology. The detailed point-by-point response to each of her/his concerns is attached below. The changes to the original submission are highlighted in blue in the revised manuscript and Supplementary Information.

Major comments:

1. The authors claim that the ‘chain walkers’ can be rapidly restored to their initial state via diffusion. Can the authors show whether diffusional forces could overcome the friction-based forces they have calculated from SMFS?

Response: We thank the reviewer for the comments. Although direct measurement of diffusional forces driven by thermal motion was not possible due to the limited force resolution (~10 pN) of our single-molecule force spectroscopy (SMFS) setup, our experiments offer insights into this aspect. Our results indicate that the friction force is significantly influenced by the velocity of the PEG chain passing through the CD ring. Specifically, an increase in pulling speed results in a higher friction force (Fig. 2d and e). By using the Bell-Evans model (Evans et al. *Biophys. J.* 1999, 76, 2439-2447) to extrapolate to zero pulling force, and assuming the diffusion process involves the CD jumping within a periodic potential dictated by the internal interaction between CD and the PEG chain (Ito et al. *J. Am. Chem. Soc.* 2019, 141, 24, 9655–9663), we determined

the diffusion speed at zero pulling forces to be 23 nm s^{-1} for α -CD and 26 nm s^{-1} for β -CD. This suggests that CD rings can diffuse rapidly in the absence of pulling forces, indicating that CD rings can overcome frictional forces for rapid diffusion when no external forces are applied to them.

During the relaxation of the hydrogel, the chain walkers (CD rings) remain stationary when friction-based forces prevent diffusion. However, as the tetra-arm PEG network undergoes entropic recoiling (depicted as the 'railway' in Fig. 1d), it draws the CD rings closer together, notably reducing their separation compared to the original distance ($P_1'P_2'$ is smaller than P_1P_2). At this point, the PEG segment linking CD rings relaxes with the restoration of the tetra-arm PEG network, alleviating any stretching force applied to the CD rings. Consequently, the chain walkers (CD rings) can rapidly return to their original positions via free diffusion, bypassing the challenge of overcoming high friction forces encountered under elevated nominal forces. The new comments have been included in the revised manuscript. (See line 130-138 on Page 6-7, line 188-201 on Page 9-10, Fig. 1 on Page 7 and Fig. 2 on Page 11 in the revised manuscript; See line 81-91 on Page 3-4 in the revised Supporting Information)

Revisions:

...During the relaxation of the hydrogel, the chain walkers (CD-PEG-CD) remain stationary when friction-based forces prevent diffusion. However, as the tetra-arm PEG network undergoes entropic recoiling (depicted as the 'railway' in Fig. 1d), it draws the CD rings closer together, notably reducing their separation compared to the original distance ($P_1'P_2'$ is smaller than P_1P_2). At this point, the PEG segment linking CD rings relaxes with the restoration of the tetra-arm PEG network, alleviating any stretching force applied to the CD rings. Consequently, the chain walkers (CD-PEG-CD) can rapidly return to their original positions via free diffusion, bypassing the challenge of overcoming high friction forces encountered under elevated nominal forces...

... In single molecule force spectroscopy (SMFS), the friction force is counteracted by

the tethering force of the linker connecting to the CD ring, resulting in a constant force plateau in the force-extension curves. The greater the tethering force, the greater the friction force (Fig. 2d and e). We can anticipate that the friction force becomes insignificant if the CD ring diffuses freely on a PEG molecule without the tethering forces exerted by the linker linked to the CD ring, akin to when the pulling speed approaches zero. Moreover, by using the Bell-Evans model⁶³ to extrapolate to zero pulling force, and assuming the diffusion process involves the CD jumping within a periodic potential dictated by the internal interaction between CD and the PEG chain⁶⁴, we determined the diffusion speed at zero pulling forces to be 23 nm s⁻¹ for α -CD and 26 nm s⁻¹ for β -CD (See Supporting Information for the calculation details). This suggests that CD rings can diffuse rapidly in the absence of pulling forces, indicating that CD rings can overcome frictional forces for rapid diffusion when no external forces are applied to them.

Determination of the diffusion speed at zero pulling forces

Based on the Bell-Evans model, the diffusion rate of CD under force (k_F) on PEG chain can be described by $k_F = k_0 \exp\left[\frac{F\Delta x}{k_B T}\right]$, where F is the friction force, k_0 is the diffusion frequency at zero force, T is the absolute temperature, k_B is the Boltzmann constant and Δx is the distance of the reaction length over which the force must be applied to reach the transition state. The diffusion rate of CD under force (k_F) can also be determined as $\frac{v}{L}$, in which v is the pulling speed and L is the length of PEG unit (~ 0.24 nm). Thus, the correlation between friction force (F) and pulling speed (v) in SMFS follows the equation: $F = \frac{k_B T}{\Delta x} \ln \frac{v}{L k_0}$. By fitting the curves of F vs. v in Fig. 2d and e, the diffusion frequencies at zero force (k_0) were obtained. At last, the diffusion speeds of CD at zero pulling forces (D) were determined as $D = k_0 L$.

Figure 1 Molecular and network engineering of hydrogels containing chain walkers. **a**, Schematic of CD sliding on end free PEG chains in conventional slide-ring hydrogels. CD slides along the PEG chain and barely restores after the stress relaxation. P_1 , P_2 , P_1' and P_2' indicate fixed positions of the polymer. The “×” mark between state i and state iii indicates that the CD rings might not be able to return to their original position. **b**, Schematic free energy landscape corresponding to different states in **a**. E_1 corresponds to the energy barrier between state ii and state iii. **c**, Schematic of the polymer network in conventional slide-ring hydrogels under deformation and relaxation. **d**, Schematic of CD sliding on end fixed PEG chains in hydrogels of this work. CD sliding can dissipate energy efficiently, and the position of the chain walkers can be rapidly restored via diffusion after the stress relaxation. P_1 , P_2 , P_1' and P_2' indicate fixed positions of the “railway” polymer. **e**, Schematic free energy landscape corresponding to different states in **d**. E_2 corresponds to the energy barrier between state i and state ii. **f**, Schematic of the hydrogel network made of covalently crosslinked four-armed PEG containing chain walkers under deformation and relaxation. CD slides and restores along PEG chains

under dynamic load bearing cycles. Blue arrows indicate the sliding directions of chain walkers along “railway” polymers.

Figure 2 SMFS and MD simulations of the friction between PEG and CD. **a**, Schematic diagram of the AFM-based SMFS experiments of the friction between PEG and CD. mPEG-SH (10 kDa) was linked to the cantilever tip via APTES and SMCC. **b**, Typical force–displacement curves for the friction between PEG and α -CD or β -CD at a pulling speed of 200 nm s^{-1} . **c**, Force histograms of frictions between PEG and α -CD or β -CD at a pulling speed of 200 nm s^{-1} . The Gaussian fitting shows average frictional forces of $26 \pm 10 \text{ pN}$ ($n = 107$) and $19 \pm 9 \text{ pN}$ ($n = 122$), respectively. **d**, **e**, Dynamic force spectroscopy experiments of sliding frictions between PEG and α -CD (**d**) or β -CD (**e**) at various pulling speeds ($200, 400, 800, 1600,$ and 3200 nm s^{-1}). Frictional forces between PEG and α -CD or β -CD are both pulling speed dependent. The dashed line indicates the linear fitting of the frictional forces and pulling speeds. The diffusion speeds at zero pulling forces were determined to be 23 nm s^{-1} for α -CD and 26 nm s^{-1} for β -CD. Values represent the mean and standard deviation ($n > 50$). **f**, Cartoon

representation of pulling a PEG ($n = 34$) through the ring of CD in a typical MD simulation. All the force fields of molecules were generated based on the general AMBER force field (GAFF) and Antechamber. **g**, Typical curves of work vs pulling distance (top) and force vs pulling distance (bottom) in α -CD and water. PEG was first pulled through a α -CD ring and then pulled in water. **h, i**, Frictional force between PEG and α -CD (**h**) or β -CD (**i**) at different pulling speeds ($1-14 \text{ m s}^{-1}$) predicted by MD simulations. Values represent the mean and standard deviation ($n = 10$).

2. As mentioned in line 107, “the conventional slide-ring hydrogels demonstrated minimal hysteresis and energy dissipation”. The minimal hysteresis is actually a sign of rapid recovery. For example, almost 100% rapid recovery (after 100th consecutive tensile cycles) and self-reinforcement is realized in previously reported slide-ring hydrogels, for which you can refer: Liu et al., *Science* 372, 1078–1081 (2021). Consequently, figure 1a (i) and (iii) showing the “un-recovery” of conventional slide-ring hydrogels is not true. The authors must properly distinguish the cyclability/recovery ability of their design and the conventional slide-ring hydrogels in the main text as well as in Figure 1?

Response: We thank the reviewer for the comments. We apologize for the misleading of the “un-recovery” of conventional slide-ring hydrogels in Fig. 1a. We totally agree with the reviewer that the slide-ring hydrogel presented by Liu et al. exhibited rapid recovery and self-reinforcement properties. With Fig. 1a-b, we meant to show that the CD rings might not be able to return to their original position due to the high energy barrier (E_I). However, the main hydrogel network can recovery, as presented by the recovered hydrogel shape and mechanical properties. Actually, the “reversible” sign in Fig. 1c also indicated that the recovery of these slide-ring hydrogels. For the damping hydrogels in this work, both the positions of CD rings and the main hydrogel network can fully recover, leading the fast restoration of energy dissipation capability. Now the label between states iii and i in Fig. 1a has been clarified into “CD recovery” to avoid the misleading on the recovery of the conventional hydrogel. We also added the corresponding comments in the caption of Fig. 1a and revised manuscript. (See line

Revisions:

...Due to the high energy barrier (E_1), the CD rings might not return to their original position while the hydrogel can recover its original shape (Fig. 1a and c). Consequently, the CD rings cannot undergo repeated and long-range frictions during the deformation cycles, resulting in minimal hysteresis and energy dissipation in these slide-ring hydrogels after the first round of cyclic deformation.^{32, 53, 57}

Figure 1 Molecular and network engineering of hydrogels containing chain walkers. **a**, Schematic of CD sliding on end free PEG chains in conventional slide-ring hydrogels. CD slides along the PEG chain and barely restores after the stress relaxation. P_1 , P_2 , P_1' and P_2' indicate fixed positions of the polymer. The “×” mark between state i and state iii indicates that the CD rings might not be able to return to their original position. **b**, Schematic free energy landscape corresponding to different states in **a**. E_1 corresponds to the energy barrier between state ii and state iii. **c**, Schematic of the polymer network

in conventional slide-ring hydrogels under deformation and relaxation. **d**, Schematic of CD sliding on end fixed PEG chains in hydrogels of this work. CD sliding can dissipate energy efficiently, and the position of the chain walkers can be rapidly restored via diffusion after the stress relaxation. P_1 , P_2 , P_1' and P_2' indicate fixed positions of the “railway” polymer. **e**, Schematic free energy landscape corresponding to different states in **d**. E_2 corresponds to the energy barrier between state i and state ii. **f**, Schematic of the hydrogel network made of covalently crosslinked four-armed PEG containing chain walkers under deformation and relaxation. CD slides and restores along PEG chains under dynamic load bearing cycles. Blue arrows indicate the sliding directions of chain walkers along “railway” polymers.

3. The authors must explain why the strain at break is improved by the ‘chain walkers’, compared to the control hydrogel (Supplementary Figure 11a)? Presumably only the modulus would be increased (by the friction of chains sliding through cyclodextrin), while the strain would still be limited by the covalent PEG crosslinks.

Response: We thank the reviewer for the comments. Even when the covalently crosslinked network breaks, the broken chains must thread out of the CD rings to detach completely. Therefore, we propose that the ‘chain walkers’ moving along the PEG chains at the break can also dissipate energy, thereby preventing crack propagation at the break point and increasing the break strain. Moreover, the ‘chain walkers’ have the potential to alter network connectivity. In conventional tetra-PEG hydrogels, not all end groups fully react to form covalent crosslinks. Some ‘chain walkers’ present on distinct tetra-PEG molecules can act as mobile crosslinks, akin to those observed in typical slide-ring hydrogels (Ito et al. *Science* 2021, 372,1078-1081; Ito et al. *Chem. Mater.* 2018, 30, 15, 5013–5019; Takeoka et al. *Nat. Commun.* 2014, 5, 5124; Ito et al. *Adv. Mater.* 2001, 13, 485-487), further contributing to an increase in break strain. We have incorporated these new insights into the revised manuscript. (See the line 274-280 on Page 14-15 in the revised manuscript)

Revisions:

...Notably, the presence of PEG-CD enhanced the fracture strain of hydrogels, likely due to that the 'chain walkers' moving along the PEG chains at the break can also dissipate energy, thereby preventing crack propagation at the break point and increasing the break strain. Moreover, the 'chain walkers' present on distinct tetra-PEG molecules can act as mobile crosslinks for the end free polymers that do not form covalent crosslinks, thus altering network connectivity akin to those observed in typical sliding hydrogels.

4. There should be further experimental comparisons performed with the control gels. How does the successive cycling and rate-dependent behavior of the control gels compare?

Response: We thank the reviewer for the comments. Following the reviewer's comment, we performed successive cycling and rate-dependent experiments on the control hydrogels. As shown in Supplementary Fig. 19, the energy dissipation of the control groups remained low and decreased to ~90% over 20 cycles of compression-relaxation. Additionally, the energy dissipation of the control hydrogels under different deformation rates (0.2%-200% min⁻¹) remained almost the same without any observed enhancement with increased deformation rates, suggesting negligible rate-dependent properties of the control hydrogels (Supplementary Fig. 16). Now we have added the new data and comments in the revised manuscript. (See line 308-310 on Page 16 and line 325-327 on Page 17 in the revised manuscript; See Supplementary Fig. 16 on Page 13 and Supplementary Fig. 19 on Page 14 in the revised Supplementary Information)

Revisions:

...In contrast, the energy dissipations of the control hydrogels remained almost the same under different deformation rates, suggesting negligible rate-dependent properties (Supplementary Fig. 16)...

...In contrast, the energy dissipations of the control groups remained low and decreased slightly over 20 cycles of compression-relaxation (Supplementary Fig. 19)...

Supplementary Figure 16. Rate-dependent performance of hydrogels without PEG-CD. **a**, Compression-relaxation curves of hydrogels without PEG-CD at different deformation rates (0.2%, 2%, 20% and 200% min⁻¹) at a strain of 50%. **b**, **c**, Energy dissipation and relative energy dissipation of hydrogels without PEG-CD at different deformation rates (0.2%, 2%, 20%, and 200% min⁻¹) at a strain of 50%. Values represent the mean and standard deviation (n = 3).

Supplementary Figure 19. Recovery performance of hydrogels without PEG-CD under consecutive compression-relaxation cycles. **a**, Consecutive compression-relaxation cycles of hydrogels without PEG-CD for 20 cycles at the deformation rate of 20% min⁻¹ without any waiting time. **b**, Normalized maximum stress and dissipated energy of hydrogels without PEG-CD in 20 cycles of continuous compression-relaxation.

5. It is unclear whether the improved cell viability is only due to a damping effect. It is well documented that viscoelasticity in hydrogels can improve cell viability via stress relaxation. The authors should perform rheological experiments to determine the difference in stress relaxation and storage/loss modulus viscoelastic properties between

the damping gel and the control PEG gels.

Response: We thank the reviewer for the comments. Following the reviewer's comments, we conducted rheological experiments on damping hydrogels and control hydrogels. As shown in Supplementary Fig. 30a-b and d-e, the storage moduli (G') of damping hydrogels and control hydrogels were almost overlapped under both frequency and strain sweeps. The loss moduli of all the hydrogels were quite small, contributing less than 1.5% to the complex moduli (G^* , defined as $\sqrt{(G' + G'')}$), suggesting an elastic behavior for all the hydrogels. Moreover, the characteristic relaxation times (τ), defined as the time when G_t reached a stable plateau, were almost the same for all hydrogels as indicated by the stress-relaxation analysis (Supplementary Fig. 30c and f). These results revealed the similar elasticity and stress-relaxation for different hydrogels, indicating that the improved viability was mostly due to the damping effect. We have now added the new data and comments in the revised manuscript. (See line 382-391 on Page 19-20 in the revised manuscript; See Supplementary Fig. 30 on Page 22 in the revised Supplementary Information)

Revisions:

...The storage/loss moduli and stress-relaxation rates of damping hydrogels and control hydrogels were almost the same, as shown by rheological analyses, revealing similar viscoelasticity and stress-relaxation for different hydrogels (Supplementary Fig. 30). These findings demonstrate that the enhanced energy dissipation in hydrogels, attributed to molecular frictions, effectively guards encapsulated stem cells against excessive mechanical damage under loading. The efficient and rapid recovery of damping hydrogels during multiple stress-relaxation cycles release mechanical load efficiently with each cycle, further ensuring effective protection of cells under continuous compression-relaxation cycles.

Supplementary Figure 30. Rheological characterization of different hydrogels. The hydrogel prepared without chain walkers was used as the control. **a, b**, Rheological measurements of hydrogels at different PEG-SH:PEG- α -CD ratios (1:1 and 1:2) under the frequency-sweep (**a**) and strain-sweep (**b**) modes. **c**, Stress relaxation analysis for hydrogels at different PEG-SH:PEG- α -CD ratios (1:1 and 1:2) based on rheology measurements. The dashed-black line corresponds to the time when G_t reached a stable plateau, which is defined as the characteristic relaxation time (τ). **d, e**, Rheological measurements of hydrogels at different PEG-SH:PEG- β -CD ratios (1:1 and 1:2) under the frequency-sweep (**d**) and strain-sweep (**e**) modes. **f**, Stress relaxation analysis for hydrogels at different PEG-SH:PEG- β -CD ratios (1:1 and 1:2) based on rheology measurements. The dashed-black line corresponds to the time when G_t reached a stable plateau, which is defined as the characteristic relaxation time (τ).

6. Is the mechanical cycling performance rate dependent? At higher rates of compression/relaxation, one would expect that the ‘chain walkers’ may not have enough time to diffuse to their low energy state between cycles?

Response: We thank the reviewer for the comments. Following the reviewer’s comments, we conducted cycling compression-relaxation experiments on damping hydrogels at higher loading rates (Supplementary Fig. 20). As predicted by the reviewer, the energy dissipation of damping hydrogels after 20 cycles of stress-relaxations at a

loading rate of $200\% \text{ min}^{-1}$ decreased slightly to $\sim 89\%$, while that at a loading rate of $20\% \text{ min}^{-1}$ was $\sim 94\%$. This can be attributed to that the ‘chain walkers’ may not fully restore their positions at high deformation rates. We have included the new data and comments in the revised manuscript. (See line 327-330 on Page 17 in the revised manuscript; See Supplementary Fig. 20 on Page 15 in the revised Supplementary Information)

Revisions:

...Moreover, the energy dissipation of damping hydrogels decreased to $\sim 89\%$ after 20 cycles of stress-relaxations at the higher loading rate ($200\% \text{ min}^{-1}$), which can be attributed to that the ‘chain walkers’ may not fully restore their positions at high deformation rates (Supplementary Fig. 20)...

Supplementary Figure 20. Energy dissipation of damping hydrogels at high deformation rates. **a**, Consecutive compression-relaxation cycles of hydrogels containing PEG- α -CD without any waiting time for 20 cycles at the deformation rate of $200\% \text{ min}^{-1}$ (PEG-SH:PEG- α -CD = 1:1). **b**, Consecutive compression-relaxation cycles of hydrogels containing PEG- β -CD without any waiting time for 20 cycles at the deformation rate of $200\% \text{ min}^{-1}$ (PEG-SH:PEG- β -CD = 1:1). **c**, **d**, Normalized energy dissipation of hydrogels containing PEG- α -CD (**c**) or PEG- β -CD (**d**) in 20 consecutive

compression-relaxation cycles at the deformation rate of $200\% \text{ min}^{-1}$ (PEG-SH:PEG - CD = 1:1).

7. Is there any permanent deformation experienced by either of the gels during this 2000 cycling experiment? If the control hydrogel is undergoing permanent deformation over cycles, this may explain the poor cell viability.

Response: We thank the reviewer for the comments. Following the reviewer's comments, we evaluated the permanent deformation of different hydrogels after 2000 cycles of compression-relaxation. As shown in Supplementary Fig. 29, the permanent deformations of damping hydrogels were lower than 5% after 2000 cycles of compression-relaxation, indicating that the cytoprotecting effect was mainly attributed to the enhanced energy dissipation of damping hydrogels. Interestingly, the permanent deformation of the control hydrogel was also lower than 5%, probably due to the elasticity of the covalently crosslinked hydrogel network. Now we have added the new data and comments in the revised manuscript. (See line 381-382 on Page 19 in the revised manuscript; See Supplementary Fig. 29 on Page 21 in the revised Supplementary Information)

Revisions:

Furthermore, the permanent deformations of different hydrogels were lower than 5% after 2000 cycles of compression-relaxation (Supplementary Fig. 29)...

Supplementary Figure 29. Mechanical performances of different hydrogels in 2000 cycles of compression-relaxation. **a**, Consecutive compression-relaxation cycles of control hydrogels without any waiting time for 2000 cycles (frequency ~ 0.5 Hz). **b**, Consecutive compression-relaxation cycles of hydrogels containing PEG- α -CD without any waiting time for 2000 cycles (PEG-SH:PEG- α -CD = 1:1, frequency ~ 0.5 Hz). **c**, Consecutive compression-relaxation cycles of hydrogels containing PEG- α -CD without any waiting time for 2000 cycles (PEG-SH:PEG- α -CD = 1:2, frequency ~ 0.5 Hz). **d**, Consecutive compression-relaxation cycles of hydrogels containing PEG- β -CD without any waiting time for 2000 cycles (PEG-SH:PEG- β -CD = 1:1, frequency ~ 0.5 Hz). **e**, Consecutive compression-relaxation cycles of hydrogels containing PEG- β -CD without any waiting time for 2000 cycles (PEG-SH:PEG- β -CD = 1:2, frequency ~ 0.5 Hz). **f**, Summarized permanent deformations of different hydrogels after 2000 cycles of compression-relaxation. Values represent the mean and standard deviation ($n = 3$).

Minor comments:

1. In Figure 1, should the direction of the deformation and relaxation arrows be swapped?

Response: We thank the reviewer for the comments. We apologize for the mislabeling of the arrows. Now we have swapped the direction of the deformation and relaxation

arrows. (See revised Fig. 1 on Page 7 in the revised manuscript)

Revisions:

Figure 1 Molecular and network engineering of hydrogels containing chain walkers. **a**, Schematic of CD sliding on end free PEG chains in conventional slide-ring hydrogels. CD slides along the PEG chain and barely restores after the stress relaxation. P_1 , P_2 , P_1' and P_2' indicate fixed positions of the polymer. The “×” mark between state i and state iii indicates that the CD rings might not be able to return to their original position. **b**, Schematic free energy landscape corresponding to different states in **a**. E_1 corresponds to the energy barrier between state ii and state iii. **c**, Schematic of the polymer network in conventional slide-ring hydrogels under deformation and relaxation. **d**, Schematic of CD sliding on end fixed PEG chains in hydrogels of this work. CD sliding can dissipate energy efficiently, and the position of the chain walkers can be rapidly restored via diffusion after the stress relaxation. P_1 , P_2 , P_1' and P_2' indicate fixed positions of the “railway” polymer. **e**, Schematic free energy landscape corresponding to different states in **d**. E_2 corresponds to the energy barrier between state i and state ii. **f**, Schematic of

the hydrogel network made of covalently crosslinked four-armed PEG containing chain walkers under deformation and relaxation. CD slides and restores along PEG chains under dynamic load bearing cycles. Blue arrows indicate the sliding directions of chain walkers along “railway” polymers.

2.The authors need to include an explanation for their energy dissipation calculation. Response: We thank the reviewer for the comments. We apologize for missing the details for the calculation of energy dissipation. The dissipated energy (E) was calculated as the area enclosed by each compression-relaxation cycle. The relative energy dissipation was determined as $\varphi = \frac{E}{E_0} \times 100\%$, in which E corresponds to the dissipated energy and E_0 is calculated as follows: $E_0 = \int_{x_0}^{x_e} \sigma(x)dx$, in which x_0 and x_e correspond to the starting strain and the maximum strain of the compression, and $\sigma(x)$ corresponds to the compression curve. Now we have included an explanation of the energy dissipation calculation in the revised manuscript. (See line 583-588 on Page 29-30 in the revised manuscript)

Revisions:

...The dissipated energy (E) was calculated as the area enclosed by each compression-relaxation cycle. The relative energy dissipation was determined as $\varphi = \frac{E}{E_0} \times 100\%$, in which E corresponds to the dissipated energy and E_0 is calculated as follows: $E_0 = \int_{x_0}^{x_e} \sigma(x)dx$, in which x_0 and x_e correspond to the starting strain and the maximum strain of the compression, and $\sigma(x)$ corresponds to the compression curve.

3.While Figure 1d seems understandable to interpret the CD-PEG-CD as a “chain walker”, the reviewer had a hard time understanding the network structure changes during deformation with Figure 1f. It would be helpful to indicate the motion direction for either the CD ring or the PEG chain. Can the authors please improve this schematic? Response: We thank the reviewer for the comments. In the revised schematic, we have

indicated the motion direction for the CD ring with blue arrows and highlighted a typical sliding of the chain walker with a red frame (Fig. 1f). Additionally, we have added black arrows below the deformed hydrogel network to indicate the deformation direction of the hydrogel (right of Fig. 1f). The improved Fig. 1 has been included in the revised manuscript. (See Fig. 1f on Page 7 in the revised manuscript)

Revisions:

Figure 1 Molecular and network engineering of hydrogels containing chain walkers. **a**, Schematic of CD sliding on end free PEG chains in conventional slide-ring hydrogels. CD slides along the PEG chain and barely restores after the stress relaxation. P_1 , P_2 , P_1' and P_2' indicate fixed positions of the polymer. The “×” mark between state i and state iii indicates that the CD rings might not be able to return to their original position. **b**, Schematic free energy landscape corresponding to different states in **a**. E_1 corresponds to the energy barrier between state ii and state iii. **c**, Schematic of the polymer network in conventional slide-ring hydrogels under deformation and relaxation. **d**, Schematic of

CD sliding on end fixed PEG chains in hydrogels of this work. CD sliding can dissipate energy efficiently, and the position of the chain walkers can be rapidly restored via diffusion after the stress relaxation. P_1 , P_2 , P_1' and P_2' indicate fixed positions of the “railway” polymer. **e**, Schematic free energy landscape corresponding to different states in **d**. E_2 corresponds to the energy barrier between state i and state ii. **f**, Schematic of the hydrogel network made of covalently crosslinked four-armed PEG containing chain walkers under deformation and relaxation. CD slides and restores along PEG chains under dynamic load bearing cycles. Blue arrows indicate the sliding directions of chain walkers along “railway” polymers.

Reviewer #2 (Remarks to the Author):

In the manuscript by Zhengyu Xu, Jiajun Lu, et al., the authors introduce a molecular-level damping mechanism into hydrogels by embedding slide-ring-based intra- and inter-chain linkers. This mechanism can achieve relatively substantial energy dissipation, as well as rapid recovery, which is attributed to the fast dynamics of bridged slide-rings on the polymer backbone. The authors further demonstrate that the hydrogels they designed can effectively shield encapsulated cells from mechanical trauma under cyclic compression, holding potential for use as biomaterials in dynamic loading circumstances. Overall, the design of this work is interesting, but the interpretation of the experimental results could be improved. I would reconsider it for publication in Nature Communications after major revision. Here are several points that might be helpful:

General Response: We thank the reviewer for the insightful comments. The reviewer has raised several important points regarding our mechanism discussion and explanation, offering valuable insights to enhance the interpretation of our experimental results. In response, we have conducted a thorough revision, incorporating new discussions and revising existing ones, as well as performing additional experiments. The detailed point-by-point response to each of her/his concerns is attached below. The changes to the original submission are highlighted in blue in the revised manuscript and Supplementary Information.

1. In lines 52-54, there is a very important concept that the authors need to clarify. The sacrificial bonds or interactions themselves do not store too much elastic energy. Instead, it is the polymer chains that store the majority of the elastic energy, which gets released upon the scission of the sacrificial bonds.

Response: We thank the reviewer for the comments. Our original statement was not accurate. We agree with the reviewer that sacrificial bonds or interactions themselves do not store significant elastic energy. Both the enthalpic energy stored by the sacrificial bonds or interaction and the elastic energy stored in the polymer chains crosslinked by these bonds are released upon the dissociation of sacrificial bonds. We have clarified

this in the revised manuscript. (See line 52-56 on Page 3 in the revised manuscript)

Revisions:

... The breaking of sacrificial bonds or interactions results in the release of both the enthalpic energy stored within these bonds and the elastic energy stored in the polymer chains connected by these bonds. This process leads to an increase in hysteresis and the apparent work needed to disrupt the internal network...

2. In lines 54-56, the statement “However, to enhance energy dissipation, the energy of these sacrificial bonds/networks needs to be substantial, which can compromise the dynamic properties of the hydrogel” is misleading. It is not the substantial energy of the sacrificial bonds that compromises the dynamic properties of the hydrogels; rather, it is the slow reformation and the inability to find the original pair that compromise the fast recovery of the gels.

Response: We thank the reviewer for the comments. We apologize for the incorrect delivery of information. As highlighted by the reviewer, the slow kinetics of the crosslinker's reformation, combined with the challenge of matching the original pairs, collectively hinder the rapid recovery of hydrogels. Typically, sacrificial bonds of higher stability enable the network to store a greater amount of elastic energy, which is then dissipated upon rupture. Generally, chemical bonds characterized by high bond energy are stable but reform at slow rates (*Angew. Chem. Int. Ed.* 2022, 61, e202206938; *J. Therm. Anal. Calorim.* 2021, 143, 3439-3445), potentially impeding the swift recovery of hydrogels. We have revised the statement following the reviewer's comments in the revised manuscript. (See line 56-59 on Page 3 in the revised manuscript)

Revisions:

... However, to enhance energy dissipation, the energy of these sacrificial bonds needs to be substantial, which can lead to generally slow reformation rates.^{37,38} The slow reformation and the inability to find the original pair in hydrogel networks compromise

the fast recovery of hydrogels.^{28,39-41} ...

3. In lines 82-84, what limits the mobility of the CD rings in slide-ring gels under stress? The slide-ring gels developed by Ito and coworkers are very resilient and can fully recover their original shapes after large deformation. The lack of an energy dissipation mechanism in the gels is due to their high elasticity under typical strain rates, not because they cannot recover. The picture the authors describe is different from my understanding of slide-ring gels. The authors should provide references for this statement.

Response: We thank the reviewer for the comments. We apologize for any confusion caused by our previous explanation regarding the mobility of CD rings in slide-ring gels. In conventional slide-ring hydrogels, the mobility of CD rings is not constrained under stress. As presented by Ito and colleagues, in slide-ring hydrogels, polymer chains can pass through CD rings, which act like pulleys to cooperatively equalize the tension of polymer chains (*Chem. Mater.* 2018, 30(15), 5013–5019; *Adv. Mater.* 2001, 13(7), 485-487). This tension equalization occurs not only within individual polymer chains but also among adjacent interlocked cross-links. This 'pulley effect' results in CD rings being positioned where tension on both sides is balanced after the initial pulling cycle, limiting their subsequent movement. In contrast, our hydrogels were designed with careful consideration of the mismatch between the lengths of the 'chain walker' legs and the polymer railway, as well as a predefined range of walker movement. These design features ensure effective friction and rapid recovery of CD positions. We have clarified these points in the revised manuscript. (See line 83-89 on Page 4-5 in the revised manuscript)

Revisions:

...In these slide-ring hydrogels, polymer chains can pass through CD rings, which act like pulleys to cooperatively equalize the tension of polymer chains.^{53,56} The tension equalization occurs not only within individual polymer chains but also among adjacent interlocked cross-links. This 'pulley effect' results in CD rings being positioned where

tension on both sides is balanced after the initial pulling cycle, limiting their subsequent movement.

4. In lines 103-106, the free energy of the typical slide-ring gels at small deformation comes from two parts: the entropy of the rings and the entropy of the chains. I am skeptical about the schematic potential profile shown in figure 1b. There should be no significant free energy difference between states i and iii. Here are two main reasons: Firstly, without external load, the chains are in their ideal state. The chain between crosslinks is always dominated by thermal fluctuation (kT). Since the number of chains is the same in states i and iii, they should have similar free energy. States i and iii might have slightly different free energy in terms of the rings, but that is trivial. Secondly, if figure 1a is accurate, slide-ring gels would have a higher loading Young's modulus than their unloading Young's modulus since modulus indicates the free energy density within the networks. This is not observed in the system developed by Ito and coworkers. I can imagine the rings might not be able to return to their original position after deformation, but it does not mean the gel cannot recover its original shape. The number of chains remains unchanged.

Response: We thank the reviewer for the comments. We agree that there should be no significant difference in free energy between states i and iii. As the reviewer pointed out, the polymer network should have similar free energy due to the same number of chains, and the difference in ring positions can slightly affect the free energy. We meant to emphasize that the ring position of CD cannot fully recover after deformation, rather than showing the changing of crosslinking density or polymer entanglements. In conventional slide-ring hydrogels, the hydrogel can recover its original shape while the rings might not be able to return to their original position.

In our work, the sliding-ring process was specially utilized to dissipate energy via molecular friction. On one hand, the hydrogel network needs to be carefully designed, and the mismatch between the lengths of the 'chain walker' legs and the track is employed to create friction during stretching. On the other hand, the PEG backbone is covalently crosslinked. As a result, the elasticity of the covalently crosslinked hydrogel

network leads to the fast recovery of 'railway' strands and retracts the CD-PEG-CD chain walkers. These two major mechanisms contribute to the enhanced amplitude and fast restoration of energy dissipation in damping hydrogels.

We have revised the schematic potential profile in Fig. 1b by tuning down the difference between states i and iii. Meanwhile, the label between states iii and i in Fig. 1a has been clarified as “CD recovery” to avoid misleading on the recovery of the hydrogel shape. We have also added corresponding comments in the caption of Fig. 1a and revised the manuscript. (See line 108-116 on Page 5-6 and Fig. 1 on Page 7 in the revised manuscript)

Revisions:

...However, after relaxation, the network reverts to a more uniform arrangement with the CD rings distributed more evenly, resulting in a new energy level which is slightly different from that in the initial state (iii of Fig. 1a-b). Due to the high energy barrier (E_I), the CD rings might not return to their original position while the hydrogel can recover its original shape (Fig. 1a and c). Consequently, the CD rings cannot undergo repeated and long-range frictions during the deformation cycles, resulting in minimal hysteresis and energy dissipation in these slide-ring hydrogels after the first round of cyclic deformation.^{32,53,57}

Figure 1 Molecular and network engineering of hydrogels containing chain walkers. **a**, Schematic of CD sliding on end free PEG chains in conventional slide-ring hydrogels. CD slides along the PEG chain and barely restores after the stress relaxation. P_1 , P_2 , P_1' and P_2' indicate fixed positions of the polymer. The “×” mark between state i and state iii indicates that the CD rings might not be able to return to their original position. **b**, Schematic free energy landscape corresponding to different states in **a**. E_1 corresponds to the energy barrier between state ii and state iii. **c**, Schematic of the polymer network in conventional slide-ring hydrogels under deformation and relaxation. **d**, Schematic of CD sliding on end fixed PEG chains in hydrogels of this work. CD sliding can dissipate energy efficiently, and the position of the chain walkers can be rapidly restored via diffusion after the stress relaxation. P_1 , P_2 , P_1' and P_2' indicate fixed positions of the “railway” polymer. **e**, Schematic free energy landscape corresponding to different states in **d**. E_2 corresponds to the energy barrier between state i and state ii. **f**, Schematic of the hydrogel network made of covalently crosslinked four-armed PEG containing chain walkers under deformation and relaxation. CD slides and restores along PEG chains

under dynamic load bearing cycles. Blue arrows indicate the sliding directions of chain walkers along “railway” polymers.

5. In Figure 1e, I do not believe state iii is unstable. There should be some activation barrier to overcome to return to state i. The barrier is just smaller than that in the case of figure 1a, and it occurs very fast under thermal fluctuation.

Response: We thank the reviewer for the comments. We agree with the reviewer that state iii should be energetically similar to state i, which is stable. The state iii should be an intermediate state during stress-relaxation, wherein the 'railway' strand has not returned to its original position, and the chain walkers remain at a relatively stable position. When the 'railway' strand almost fully returns, the chain walkers also return to their original position on the 'railway' rapidly due to a smaller barrier for conversion from state iii to state i than that in the case of Fig. 1a. Now we have moved state iii to a relative stable potential trough in Fig. 1e and the barrier relative to state i (E_2) was smaller than that of Fig. 1b (E_1). The new Fig. 1 and comments have been included in the revised manuscript. (See line 139-141 and revised Fig. 1 on Page 7 in the revised manuscript)

Revisions:

...Additionally, the energy barrier between states i and iii in the case of Fig. 1e (E_2) is smaller than that of Fig. 1b (E_1), facilitating the rapid recovery rate of chain walkers under thermal fluctuation...

Figure 1 Molecular and network engineering of hydrogels containing chain walkers. **a**, Schematic of CD sliding on end free PEG chains in conventional slide-ring hydrogels. CD slides along the PEG chain and barely restores after the stress relaxation. P_1 , P_2 , P_1' and P_2' indicate fixed positions of the polymer. The “×” mark between state i and state iii indicates that the CD rings might not be able to return to their original position. **b**, Schematic free energy landscape corresponding to different states in **a**. E_1 corresponds to the energy barrier between state ii and state iii. **c**, Schematic of the polymer network in conventional slide-ring hydrogels under deformation and relaxation. **d**, Schematic of CD sliding on end fixed PEG chains in hydrogels of this work. CD sliding can dissipate energy efficiently, and the position of the chain walkers can be rapidly restored via diffusion after the stress relaxation. P_1 , P_2 , P_1' and P_2' indicate fixed positions of the “railway” polymer. **e**, Schematic free energy landscape corresponding to different states in **d**. E_2 corresponds to the energy barrier between state i and state ii. **f**, Schematic of the hydrogel network made of covalently crosslinked four-armed PEG containing chain walkers under deformation and relaxation. CD slides and restores along PEG chains

under dynamic load bearing cycles. Blue arrows indicate the sliding directions of chain walkers along “railway” polymers.

6. Based on the gel synthesis procedure described by the authors, there should be both intra- and inter-chain slide-ring linkers. Figure 1d shows the former, while Figure 1e shows the latter. Although they might not be significantly different, it would be beneficial if the authors could clarify that they are not the same.

Response: We thank the reviewer for the comments. We apologize for missing the crosslinking on the middle of polymer chain in Fig. 1d. The medium of Fig. 1d should be the crosslinking point of a four-armed PEG, in which case Fig. 1d also corresponds the inter-chain slide-ring linker. Now we have added the crosslinking in Fig. 1d in the revised manuscript. (See revised Fig. 1d on Page 7 in the revised manuscript)

Revisions:

Figure 1 Molecular and network engineering of hydrogels containing chain walkers. **a,**

Schematic of CD sliding on end free PEG chains in conventional slide-ring hydrogels. CD slides along the PEG chain and barely restores after the stress relaxation. P_1 , P_2 , P_1' and P_2' indicate fixed positions of the polymer. The “×” mark between state i and state iii indicates that the CD rings might not be able to return to their original position. **b**, Schematic free energy landscape corresponding to different states in **a**. E_1 corresponds to the energy barrier between state ii and state iii. **c**, Schematic of the polymer network in conventional slide-ring hydrogels under deformation and relaxation. **d**, Schematic of CD sliding on end fixed PEG chains in hydrogels of this work. CD sliding can dissipate energy efficiently, and the position of the chain walkers can be rapidly restored via diffusion after the stress relaxation. P_1 , P_2 , P_1' and P_2' indicate fixed positions of the “railway” polymer. **e**, Schematic free energy landscape corresponding to different states in **d**. E_2 corresponds to the energy barrier between state i and state ii. **f**, Schematic of the hydrogel network made of covalently crosslinked four-armed PEG containing chain walkers under deformation and relaxation. CD slides and restores along PEG chains under dynamic load bearing cycles. Blue arrows indicate the sliding directions of chain walkers along “railway” polymers.

7. In the section on SMFS, is there a difference in the success rate of threading a PEG chain through alpha-CD versus beta-CD? This is a minor point, but it piques my curiosity.

Response: We thank the reviewer for the comments. In SMFS, the densities of CD molecules modified on the substrates were kept extremely low to prevent the simultaneous picking up of multiple molecules. Consequently, the success rate of threading a PEG chain through CD rings is not governed by threading kinetics but rather by the number of available CD rings and PEG chains in the contact region between the cantilever tip and the substrate. As summarized in Supplementary Fig. 2, the pick-up rates of SMFS for α -CD and β -CD were 1.64% and 1.71%, respectively. This suggests that there is no significant difference in the success rates of threading a PEG chain through different CD rings in SMFS. We have added the new comment in the revised manuscript. (See line 177-179 on Page 9 in the revised manuscript; See Supplementary

Fig. 2 on Page 5 in the revised Supplementary Information)

Revisions:

...Moreover, no obvious difference between pickup rates of α -CD and β -CD were observed, indicating the similar success rates of threading a PEG chain through different CD rings in SMFS experiments...

Supplementary Figure 2. SMFS on glass substrates without CD. **a**, Schematic diagram of the AFM-based SMFS experiments on glass substrates without CD. mPEG-SH (10 kDa) was linked to the cantilever tip via APTES and SMCC. **b**, Typical force–displacement curves of SMFS on glass substrates without CD at a pulling speed of 200 nm s⁻¹. In most traces, no force plateau was observed. **c**, Pickup rates (the success rate to obtain force-extension curves showing the force plateau) of SMFS on substrates modified with and without CD at a pulling speed of 200 nm s⁻¹. Values represent the mean and standard deviation (n = 5 independent experiments). *P* values, two-tailed Student’s *t* test.

8. In figure 3a, the unloading curve of the 1:2 ratio is lower than that of the control, suggesting that the elastically active structure of the 1:2 gel is likely less than that of the control. There could be more defects in the 1:2 gel than in the control, so some part of the hysteresis of the 1:2 gel probably has contributions from dangling defects. It is probably beyond the scope of this paper, but it would be truly helpful if the authors could confirm that the elastically active structures of the different gels are the same, making the dissipation mechanism they proposed more convincing.

Response: We thank the reviewer for the comments. Following the reviewer's comment, we have evaluated the viscoelasticity of different hydrogels using rheological tests. As shown in Supplementary Fig. 30a-b and d-e, the storage moduli (G') of damping hydrogels and control hydrogels were almost overlapped either under both frequency and strain sweeps, suggesting the same elastically active structures for different hydrogels. The loss moduli (G'') of all the hydrogels were quite small, contributing less than 1.5% to the complex moduli (G^* , defined as $\sqrt{(G' + G'')}$), indicating an almost completely elastic response for all hydrogels. These results further demonstrated that the enhanced energy dissipation in hydrogels was mainly attributed to molecular frictions. We have added the new comments and data in the revised manuscript. (See line 382-388 on Page 19-20 in the revised manuscript; See Supplementary Fig. 30 on Page 22 in the revised Supplementary Information)

Revisions:

...The storage/loss moduli and stress-relaxation rates of damping hydrogels and control hydrogels were almost the same, as shown by rheological analyses, revealing similar viscoelasticity and stress-relaxation for different hydrogels (Supplementary Fig. 30). These findings demonstrate that the enhanced energy dissipation in hydrogels, attributed to molecular frictions, effectively guards encapsulated stem cells against excessive mechanical damage under loading...

Supplementary Figure 30. Rheological characterization of different hydrogels. The hydrogel prepared without chain walkers was used as the control. **a, b**, Rheological measurements of hydrogels at different PEG-SH:PEG- α -CD ratios (1:1 and 1:2) under the frequency-sweep (**a**) and strain-sweep (**b**) modes. **c**, Stress relaxation analysis for hydrogels at different PEG-SH:PEG- α -CD ratios (1:1 and 1:2) based on rheology measurements. The dashed-black line corresponds to the time when G_t reached a stable plateau, which is defined as the characteristic relaxation time (τ). **d, e**, Rheological measurements of hydrogels at different PEG-SH:PEG- β -CD ratios (1:1 and 1:2) under the frequency-sweep (**d**) and strain-sweep (**e**) modes. **f**, Stress relaxation analysis for hydrogels at different PEG-SH:PEG- β -CD ratios (1:1 and 1:2) based on rheology measurements. The dashed-black line corresponds to the time when G_t reached a stable plateau, which is defined as the characteristic relaxation time (τ).

9. In figures 3d and h, the authors should indicate the loading rate.

Response: We thank the reviewer for their comments and apologize for the omission of the loading rate in Fig. 3d and h, which was $20\% \text{ min}^{-1}$. We have now added this information to the caption. (See the revised caption of Fig. 3d and h on Page 17-18 in the revised manuscript)

Revisions:

Figure 3 Energy dissipation and fast recovery of hydrogels based on the molecular friction of chain walkers. **a**, Typical compression-relaxation curves of hydrogels at different PEG-SH:PEG- α -CD ratios (1:1 and 1:2) at a strain of 50%. The hydrogel prepared without chain walkers was used as the control. **b**, Compression-relaxation curves of hydrogels at various strains (PEG-SH:PEG- α -CD = 1:1). **c**, Compression-relaxation curves of hydrogels at different deformation rates (0.2% min⁻¹, 2% min⁻¹, 20% min⁻¹ and 200% min⁻¹) at a PEG-SH:PEG- α -CD ratio of 1:1 (strain ~50%). **d**, Consecutive compression-relaxation cycles of hydrogels containing PEG- α -CD without any waiting time for 20 cycles at the deformation rate of 20% min⁻¹ (PEG-SH:PEG- α -CD = 1:1). **e**, Typical compression-relaxation curves of hydrogels at different PEG-SH:PEG- β -CD ratios (1:1 and 1:2) at a strain of 50%. The hydrogel prepared without chain walkers was used as the control. **f**, Compression-relaxation curves of hydrogels at various strains (PEG-SH:PEG- β -CD = 1:1). **g**, Compression-relaxation curves of hydrogels at different deformation rates (0.2% min⁻¹, 2% min⁻¹, 20% min⁻¹ and 200% min⁻¹) at a PEG-SH:PEG- β -CD ratio of 1:1 (strain ~50%). **h**, Consecutive compression-relaxation cycles of hydrogels containing PEG- β -CD without any waiting time for 20 cycles at the deformation rate of 20% min⁻¹ (PEG-SH:PEG- β -CD = 1:1).

10. It is not very clear to me how the fast recovery of the gel protects cells under dynamic loading. In reference 59, energy dissipation offers protection to encapsulated cells, but the role of fast dynamics is a bit vague. Can the authors explain this in more detail? Can they clarify why their design is better than that in reference 59?

Response: We thank the reviewer for the comments. We need to clarify that it is energy dissipation, rather than the fast recovery of hydrogels, that protects cells under dynamic loading. Both our hydrogel utilizing molecular friction and the one using rupture of host-guest interactions, as referenced in 66 (originally Ref. 59), ensure effective protection of encapsulated cells through each compression-relaxation cycle. The advantage of our hydrogel lies in its efficient and rapid recovery during multiple stress-relaxation cycles, releasing mechanical load efficiently with each cycle. However, in

reference 66, the recovery is constrained by the reformation of CD-adamantane host-guest interactions, potentially limiting efficiency under high-frequency mechanical load. Molecular friction does not require the rupture and reformation of chemical bonds, preserving the intrinsic properties of hydrogels. We anticipate that this design can decouple energy dissipation from strength, presenting a novel mechanism for enhancing energy dissipation alongside traditional methods like tuning polymer concentrations and crosslinking types. We have now included these discussions in the revised manuscript. (See line 385-391 on Page 20 and line 443-448 on Page 23 in the revised manuscript)

Revisions:

...These findings demonstrate that the enhanced energy dissipation in hydrogels, attributed to molecular frictions, effectively guards encapsulated stem cells against excessive mechanical damage under loading. The efficient and rapid recovery of damping hydrogels during multiple stress-relaxation cycles release mechanical load efficiently with each cycle, further ensuring effective protection of cells under continuous compression-relaxation cycles.

...Furthermore, the energy dissipation in our design was mainly attributed to the molecular friction. Unlike in dynamic bonding systems, where the reformation of sacrificial crosslinking can constrain the recovery and limit efficiency under high-frequency mechanical load, molecular friction does not involve the rupture and reformation of chemical bonds. This holds the advantage of not affecting the intrinsic properties of hydrogels.

11. In the section on the preparation of hydrogel, the authors state that the gels were prepared at 7 mM or 2 mM. I wonder which concentration was used for figure 3? The molecular weight of the tetra PEG is 20 kDa, and 2 mM (40 mg/mL) is just at the overlap concentration. How well is the gel formed under this condition?

Response: We thank the reviewer for the comments. We apologize for the unclear

station in the caption of Fig. 3. The concentration used for Fig. 3 was 7 mM. As shown in Supplementary Fig. 28a-b, the precursors at the concentration of 2 mM can also form stable hydrogels with fixed geometry and microporous structures, similar to those at the concentration of 7 mM. However, the mechanical strength of these hydrogels was much weaker than those at higher concentrations with the moduli of ~8-9 kPa (Supplementary Fig. 28c). Now we have added the new data and comments in the revised manuscript. (See line 377-379 on Page 19 in the revised manuscript; See Supplementary Fig. 28 on Page 20 in the revised Supplementary Information)

Revisions:

...Notably, hydrogels at lower solid contents also exhibited stable geometry and microporous structures, despite the much weaker mechanical strength (Supplementary Fig. 28).

Supplementary Figure 28. Characterization of hydrogels at low PEG concentrations ($C_{\text{PEG-Mal}} = C_{\text{PEG-SH}} = 20 \text{ mg mL}^{-1}$). **a, b**, Optical (**a**) and SEM (**b**) images of different hydrogels at low PEG concentrations ($C_{\text{PEG-Mal}} = C_{\text{PEG-SH}} = 20 \text{ mg mL}^{-1}$). **c**, Stress–strain curves of different hydrogels under compression at low PEG concentrations ($C_{\text{PEG-Mal}} = C_{\text{PEG-SH}} = 20 \text{ mg mL}^{-1}$).

12. I wonder if the authors can make a PEG gel using the same procedure but use only CD molecules instead of PEG-CD? A network with CD rings on the PEG backbone but without the bridging PEG chain would be a better control compared to the current one.

Response: We thank the reviewer for the comments. Following the reviewer's comments, we prepared hydrogels with CD rings on the PEG backbone but without the bridging PEG chain. The concentrations of CD monomers used in the preparation were twice those of CD-PEG-CD, as each CD-PEG-CD (chain walker) contains two CD rings. As shown in Supplementary Fig. 14a-c, the fracture moduli and fracture strains of hydrogels with and without CD rings on the PEG backbone were almost the same, indicating that the mechanical strength was not affected by CD molecules. Moreover, the compression-relaxation curves of hydrogels with CD monomers overlapped almost perfectly with those of hydrogels without CD. The calculated energy dissipations of all the hydrogels remained almost the same and low, indicating that the introduction of CD monomers into the hydrogel network cannot enhance the energy dissipation of hydrogels. We have now added the new data and comments in the revised manuscript. (See line 296-300 on Page 15 and line 555-560 on Page 28 in the revised manuscript; See Supplementary Fig. 14 on Page 12 in the revised Supplementary Information)

Revision:

...Moreover, hydrogels containing CD monomers that were not linked via the PEG segment were also prepared for comparison. As shown in Supplementary Fig. 14, the mechanical strength and energy dissipation of these hydrogels were almost the same as those of control hydrogels, suggesting that introducing CD monomers cannot affect energy dissipation of hydrogels.

...For the preparation of hydrogels containing CD monomers, CD was dissolved in 4-armed PEG-SH solutions (7 mM) in varying proportions (PEG-SH: CD = 1:2 or 1:4), and the solutions were stored at room temperature for 4 hours to allow the PEG chain to thread through the ring. Then, the resulting solution was mixed with 4-armed PEG-

Mal solution (7 mM) at a volume ratio of 1:1. Transparent hydrogels formed after mixing and dialyzed in ddH₂O for 24 h to allow swelling equilibrium...

Supplementary Figure 14. Mechanical properties of hydrogels prepared with CD monomers that were not linked via the PEG segment. The concentrations of CD monomers used in the preparation were twice those of CD-PEG-CD, as each CD-PEG-CD (chain walker) contains two CD rings. Hydrogels without CD were used as control. **a**, Typical stress–strain curves of hydrogels at different PEG-SH:CD ratios (1:2 and 1:4). **b, c**, Summarized moduli (**b**) and fracture strains (**c**) of hydrogels at different PEG-SH:CD ratios (1:2 and 1:4). Values represent the mean and standard deviation (n = 3). **d**, Typical compression-relaxation curves of hydrogels at different PEG-SH:CD ratios (1:2 and 1:4) at a strain of 50%. **e, f**, Summarized energy dissipation (**e**) and relative energy dissipation (**f**) of hydrogels at different PEG-SH:CD ratios (1:2 and 1:4). Values represent the mean and standard deviation (n = 3).

Reviewer #3 (Remarks to the Author):

The manuscript presents an innovative approach to designing hydrogels for swift mechanical energy dissipation with a friction-based damping mechanism. By introducing molecular friction into the hydrogel matrix, the efficient energy dissipation and rapid recovery were achieved in a same hydrogel. The authors highlighted the potential applications of such hydrogels in impact protection and shock absorption such as shielding encapsulated cells during cyclic compression-relaxation. The authors explored the effective energy dissipation based on the slide-ring friction, which is really new and often overlooked in conventional hydrogels. They also emphasize the design principle of hydrogel network that facilitates the motion of a 'chain walker', leading to the efficient molecular friction.

Overall, the research is well-designed, and the experiments and simulations are carefully executed, supporting the conclusions drawn. It can provide significant insights into designing the mechanical energy dissipation of hydrogels, potentially impacting various practical applications in biomechanics and related fields. I recommend the publication of this manuscript after a few minor revisions.

General Response: We thank the reviewer for the positive comments. In order to address her/his concerns, we have conducted a substantial revision with new data, analyses and discussions. The detailed point-by-point response to each of her/his concerns is attached below. The changes to the original submission are highlighted in blue in the revised manuscript and Supplementary Information.

1. As depicted in Figure 1, it seems to me that the CD ring can slide along the PEG chain during the initial stretching. Considering that the free energy level at the initial state is higher than the final state, could the first round of deformation also induce ring sliding, and could this process dissipate energy to some extent? It might be helpful for the authors to provide clarification on this point.

Response: We thank the reviewer for the comments. We agree with the reviewer's comment that the first round of deformation also induces moderate ring sliding. In conventional slide-ring hydrogels, the ring location can be tuned under stress to achieve

a relatively stable state, which might impede subsequent frictions, resulting in moderate friction-based hysteresis primarily in the first round of cyclic deformation. This phenomenon has also been observed in previous literature (*Science* 2021, 372, 1078-1081; *Chem. Mater.* 2018, 30, 5013-5019; *Chem* 2023, 9, 3515-3531). We have now clarified this point in the revised manuscript. (See line 113-116 on Page 6 in the revised manuscript)

Revisions:

...Consequently, the CD rings cannot undergo repeated and long-range frictions during the deformation cycles, resulting in minimal hysteresis and energy dissipation in these slide-ring hydrogels after the first round of cyclic deformation.^{32, 53, 57}

2. In the MD simulation of sliding friction and molecular interactions, the frictional forces were observed to be much higher than those measured in SMFS. The authors attributed this difference to the significantly greater pulling speeds used in simulations. It would be beneficial for the authors to include a detailed discussion on this point, supported by relevant references.

Response: We thank the reviewer for the comments. According to the Bell-Evans model (*Science* 1978, 200, 618; *Biophys J.* 1999, 76, 2439-47; *Nat. Commun.* 2020, 11(1), 3895), the correlation between the most probable rupture force (F) and loading rate (r) can be described as equation (1).

$$F = \frac{k_B T}{\Delta x} \ln \left(\frac{\Delta x}{k_{off} k_B T} \right) + \frac{k_B T}{\Delta x} \ln(r) \quad (1)$$

where k_B is the Boltzmann constant, T is the absolute temperature, k_{off} is the dissociation rates and Δx is the potential width. The loading rate (r) is defined as $S \cdot v$, in which S is the slope of the force-extension curve prior to the rupture event, and v is the pulling speed. As a result, the rupture force (F) is positively correlated to the pulling speed during the SMFS experiments. Considering that the pulling speeds in MD simulations (1-14 m s⁻¹) are at least 312500-folds higher than those in the SMFS experiments (200 × 10⁻⁹-3200 × 10⁻⁹ m s⁻¹), the calculated frictional forces were much

higher than those in SMFS experiments. Now we have included the new discussions in the revised manuscript. (See line 233-238 on Page 12-13 in the revised manuscript)

Revisions:

...Noting that the rupture force (F) is positively correlated to the pulling speeds according to the Bell-Evans model^{59,63,65}, in which the correlation between the most probable rupture force (F) and loading rate (r) can be described as $F = \frac{k_B T}{\Delta x} \ln\left(\frac{\Delta x}{k_{off} k_B T}\right) + \frac{k_B T}{\Delta x} \ln(r)$. As a result, the calculated frictional forces in MD simulations were much higher than those in SMFS experiments due to the much higher pulling speed...

3. Would the introduction of CD-PEG-CD affect the initial bulk properties of the hydrogel, such as pore size? Since the authors have stated that the hydrogels exhibited similar porous microstructures, it would be helpful to provide a detailed distribution of the mesh size.

Response: We thank the reviewer for the comments. Following the reviewer's suggestion, we analyzed the mesh size distribution of hydrogels. As shown in Supplementary Fig. 10f, the mesh size of different hydrogels at the same solid contents remained almost the same, with mesh sizes located at ~1-2 μm , respectively. We have now added the new data on the mesh size distribution in the revised manuscript. (See line 267-269 on Page 14 in the revised manuscript; See Supplementary Fig. 10 on Page 9 in the revised Supplementary Information)

Revisions:

...All hydrogels showed similar swelling ratios, water contents, porous microstructures and mesh sizes (Supplementary Figs. 9-10).

Supplementary Figure 10. SEM images and mesh sizes of different hydrogels. **a**, SEM images of hydrogels at different PEG-SH:PEG-CD ratios (1:1 and 1:2). The hydrogel prepared without PEG-CD was used as a control. **b**, Average mesh sizes of different hydrogels. Values represent the mean and standard deviation ($n > 50$).

4. The mechanical stresses observed in hydrogels containing PEG-CD at high strains, such as 50% or 60%, are greater than those in hydrogels without PEG-CD. What could be the reason for this difference? Does it suggest that the CD dimer's chain walker may also function as a crosslinking agent after being stretched under significant deformations?

Response: We thank the reviewer for the comments. We agree with the reviewer's point that the higher mechanical stress of hydrogel containing PEG-CD at high strains can be attributed to that the chain walkers may act as crosslinking agent after being stretched to certain deformations. As indicated by the hydrogel network structures, the chain walkers can slide along the network during the deformation of hydrogels. When the hydrogel network is subjected to high strains, parts of the chain walkers may slide to a stable state and become trapped by the covalent crosslinking points. As a result, the chain walkers can undergo stress in hydrogels and act as fixed crosslinking points, leading to increased stress at high strains. Now we have added the new comments in the revised manuscript. (See line 280- 283 on Page 15 in the revised manuscript)

Revisions:

...At high strains, the mechanical stresses in hydrogels containing PEG-CD are greater than those in hydrogels without PEG-CD. This difference can be attributed to that parts of the chain walkers may act as fixed crosslinking points after sliding to stable states during deformations...

5. In the demonstration of protecting hMSCs from damage and loss of stemness under dynamic loading, the labels on the figures should specify α -CD or β -CD for clarity, rather than just α or β . Additionally, it seems that there were dead cells in hydrogels that did not undergo cyclic compression-relaxation. What might be the reason for this observation? Could it be due to internal stress from the hydrogel network during swelling, or could it be related to the cytotoxicity of the hydrogel? To address this question, it is recommended to include a cytotoxicity experiment.

Response: We thank the reviewer for the comments. Following the reviewer's suggestion, we have evaluated the cytotoxicity of different hydrogels by culturing cells in the presence of hydrogels instead of encapsulating cells inside hydrogels. As shown in Supplementary Fig. S22, the cell viabilities were higher than 95% in the presence of different hydrogels and nearly no dead cells were observed, indicating that hydrogels were not toxic to cells. Thus, the dead cells in hydrogels without undergoing cyclic compression-relaxation were supposed to be caused by the internal stress from the hydrogel network during swelling. We also revised the label on Supplementary Fig. 31 following the reviewer's suggestion. Now we have included the new data and comments in the revised manuscript. (See line 365-368 on Page 19 in the revised manuscript; See Supplementary Fig. 22 on Page 16 and Supplementary Fig. 31 on Page 23 in the revised Supplementary Information)

Revisions:

...The presence of dead cells in hydrogels that did not undergo cyclic compression-relaxation can be attributed to internal stress from the hydrogel network during swelling,

rather than the cytotoxicity of the hydrogels, which was found to be negligible (Supplementary Fig. 22).

Supplementary Figure 22. Cell viabilities of hMSCs cultured on different hydrogels for 24 hours. Cells cultured on cell culture plates were used as control groups. **a**, Live/dead staining of hMSCs on different hydrogels after being cultured for 24 hours. Green represents live cells with high enzymatic activity indicated by calcein-AM. The red color of PI shows dead cells with compromised membranes. **b**, Viabilities determined by live/dead staining of hMSCs on different hydrogels after culture for 24 hours. Values represent the mean and the standard deviation ($n = 5$).

Supplementary Figure 31. Stemness of hMSCs in hydrogels ($C_{\text{PEG-Mal}} = C_{\text{PEG-SH}} = 70 \text{ mg mL}^{-1}$) after 2000 cycles of compression-relaxation (strain $\sim 60\%$). A space of $1272 \mu\text{m} \times 1272 \mu\text{m} \times 300 \mu\text{m}$ was scanned, and the projected image in the Z-axis direction is shown. Immunofluorescence staining of specific markers for stemness maintenance, Oct4 (green) and Sox2 (red), was used to evaluate the stemness of hMSCs. Cell nuclei are indicated by DAPI (blue).

6. In the images of Figure 4f, there are some instances of mismatched cell staining and

noisy points. It would be preferable to replace these images with versions featuring clear backgrounds.

Response: We thank the reviewer for the comments. Now we have repeated the evaluations of stemness loss under dynamic loading and replace the images with versions featuring clear backgrounds. The new images have been included in the revised manuscript. (See Fig. 4f on Page 21 in the revised manuscript)

Revisions:

Figure 4 Damping hydrogels protect hMSCs from damage and loss of stemness under dynamic loading. **a**, Illustration of the protection of hMSCs in hydrogels under cyclic

compression-relaxation. **b, c**, Optical images (**b**) and strain signals (**c**) for cyclic compression-relaxation of hydrogels with hMSCs encapsulated inside. Conditions: frequency ~ 0.5 Hz, cycle number ~ 2000 , strain $\sim 30\%$ or 60% . **d**, 3D reconstructions of live/dead cell staining in different hydrogels using laser confocal fluorescence microscopy (LCFM) after 2000 cycles of compression-relaxation (strain: 60%). Cells were stained using calcein-AM (green) and propidium iodide (PI) (red). The size of the scanning space was $1272 \mu\text{m} \times 1272 \mu\text{m} \times 300 \mu\text{m}$. **e**, Cell viabilities of hMSCs in different hydrogels after cyclic compression-relaxation. Values represent the mean and standard deviation ($n = 3$). **f**, Stemness of hMSCs in hydrogels ($C_{\text{PEG-Mal}} = C_{\text{PEG-SH}} = 70 \text{ mg mL}^{-1}$) after cyclic compression-relaxation (strain: 60%). A space of $1272 \mu\text{m} \times 1272 \mu\text{m} \times 300 \mu\text{m}$ was scanned, and the projected image in the Z-axis direction is shown. Specific markers for stemness maintenance, Oct4 (green) and Sox2 (red), were stained. Cell nuclei are indicated by DAPI (blue). **g, h**, Normalized intensities of Oct4 (**g**) and Sox2 (**h**) for hMSCs in hydrogels after cyclic compression-relaxation (strain: 60%). The intensity of DAPI was set as 100% . Values represent the mean and standard deviation ($n = 3$).

REVIEWER COMMENTS

Reviewer #1 (Remarks to the Author):

I believe the authors have made extensive revisions, answering and addressing the majority of points raised in a conclusive manner. One final query, however, is regarding the additional stress-relaxation half time experiments that have been performed. How did the authors obtain this data? The stress relaxation appears to be abnormally fast for a hydrogel containing covalent crosslinks. Can the authors please explain this and contrast this finding to other reported systems?

Reviewer #2 (Remarks to the Author):

Overall, the authors did a commendable job addressing the reviewers' comments, but I have several points for the authors to consider.

1. Compared to other damping mechanisms, such as physical bonds, the mechanism developed based on slide-ring friction is intriguing. A crucial concept in the current manuscript is the use of the fast dynamics of slide rings to achieve rapid damping and recovery. However, comparing the developed gels with conventional slide-ring gels does not clarify this point. Conventional slide-ring gels exhibit superior mechanical properties (being highly resilient and stretchable) because the load along the polymer chain can be redistributed due to the movement of slide rings. These gels lack an intentionally introduced damping mechanism. The authors might compare their design to other damping mechanisms, such as physical bonds, to highlight their design's advantages.
2. It remains unclear why CD recovery is important for conventional slide-ring gels, as CD recovery is not crucial for achieving their exceptional mechanical properties.
3. The authors revised Figure 1 to show all slide-ring linkers bridging two network chains. However, the formation of intra-chain slide-ring linkers during gelation is also possible (original Figure 1d). The authors should discuss the potential impact of intra-chain linkers on mechanical properties.
4. Complex modulus is defined as $G^* = G' + iG''$, or $G^* = \sqrt{(G')^2 + (G'')^2}$. Not $G^* = \sqrt{G' + G''}$.

Reviewer #3 (Remarks to the Author):

The authors have addressed all the reviewers' concerns, and it is recommended for publication in Nature Communications.

Reviewer #1 (Remarks to the Author):

I believe the authors have made extensive revisions, answering and addressing the majority of points raised in a conclusive manner. One final query, however, is regarding the additional stress-relaxation half time experiments that have been performed. How did the authors obtain this data? The stress relaxation appears to be abnormally fast for a hydrogel containing covalent crosslinks. Can the authors please explain this and contrast this finding to other reported systems?

Response: We appreciate the reviewer's positive feedback on our revised manuscript and the new inquiry regarding the unusually rapid stress relaxation observed. We have identified that this may have been an artifact caused by the overshooting during the engagement of the rheometer probe with the hydrogel. The stress-relaxation experiments were conducted using a rheometer (DHR-2, TA, USA). A cylindrical hydrogel (diameter ~ 8 mm, thickness ~ 3 mm) was carefully placed on the rheometer plate, and an 8 mm parallel plate was lowered to compress the hydrogel to a thickness of 2.7 mm. Stress relaxation was then measured at a strain amplitude of 20% in the stress-relaxation mode.

During this measurement, the probe quickly moved onto the hydrogel and stopped at a pre-set strain to record the stress-relaxation. The overshooting of the probe led to mechanical perturbation, contributing to signals that did not represent true stress relaxation of the hydrogel network. Our previous interpretation incorrectly attributed these signals to the intrinsic relaxation properties of the hydrogel. Only the stress relaxation profiles ($G(t)$) measured after this phase proved reliable, showing consistent absence of notable relaxation for all three tested hydrogels within the 1-100 second timeframe. This absence of stress relaxation for covalently crosslinked hydrogels aligns with findings from previously reported systems (Zhigang Suo et al., *Journal of Applied Physics*, 2010, 107, 063509; David J. Mooney et al., *Nat. Mater.*, 2016, 15, 326–334; Ovijit Chaudhuri et al., *PNAS*, 2010, 107, 063509; Yan Xia et al., *Adv. Mater.*, 2021, 33(51), 2104460). We have now updated the manuscript to incorporate these new

insights and have omitted previous comments about the characteristic relaxation time (τ) in the revised legend of Supplementary Fig. 30. Additional references related to hydrogel stress-relaxation have been cited in the revised manuscript (See lines 384-390 on Pages 19-20 and references No. 67-70; see also the legend of Supplementary Fig. 30 on Page 22 in the revised Supplementary Information).

Revisions:

The loss moduli (G'') were significantly smaller than the storage moduli (G'), and no notable stress relaxation was observed for both damping and control hydrogels. This suggests that the mechanical response of the hydrogels predominantly originates from the covalently crosslinked network⁶⁷⁻⁷⁰, as shown in Supplementary Fig. 30. This contrasts sharply with the behavior of physically crosslinked networks, which exhibit noticeable stress relaxation due to the rupture of physical crosslinks.

Rheological Measurements

The columned hydrogels (diameter \sim 8 mm, thickness \sim 3 mm) were carefully transferred to the rheometer plate of a rheometer (DHR-2, TA, USA) prior to the measurement. The rheology experiments at the frequency-sweep mode were carried out with a frequency of 0.01 rad s⁻¹ to 100 rad s⁻¹ at 1% strain (gap: 2.7 mm). The rheology experiments at the strain-sweep mode were carried out with a strain of 0.1 to 100 % at 1 Hz (gap: 2.7 mm). The rheological stress-relaxation experiments were performed using the stress relaxation mode at a strain amplitude of 20% (gap: 2.7 mm). The temperature was 25 °C for all rheological measurements.

Supplementary Figure 30. Rheological characterization of different hydrogels. The hydrogel prepared without chain walkers was used as the control. **a, b**, Rheological measurements of hydrogels at different PEG-SH:PEG- α -CD ratios (1:1 and 1:2) under the frequency-sweep (**a**) and strain-sweep (**b**) modes. **c**, Stress relaxation analysis for hydrogels at different PEG-SH:PEG- α -CD ratios (1:1 and 1:2) based on rheology measurements. **d, e**, Rheological measurements of hydrogels at different PEG-SH:PEG- β -CD ratios (1:1 and 1:2) under the frequency-sweep (**d**) and strain-sweep (**e**) modes. **f**, Stress relaxation analysis for hydrogels at different PEG-SH:PEG- β -CD ratios (1:1 and 1:2) based on rheology measurements. For **c** and **f**, the rapid stress relaxation observed within the initial second is likely due to artifacts caused by the overshooting of the test probe during engagement with the hydrogels. This overshooting led to mechanical perturbations, resulting in signals that do not reflect the true stress relaxation of the hydrogel network. Only the stress relaxation profiles ($G(t)$) measured after this initial phase are reliable, consistently showing no significant relaxation for all three hydrogels tested within the 1-100 second timeframe.

Reviewer #2 (Remarks to the Author):

[Note from the Editor: Please also see attached PDF]

Overall, the authors did a commendable job addressing the reviewers' comments, but I have several points for the authors to consider.

Response: We appreciate the reviewer's positive comments. To address the concerns raised, we have prepared a detailed point-by-point response, which is attached below. Changes made to the manuscript are highlighted in blue in the revised version.

1. Compared to other damping mechanisms, such as physical bonds, the mechanism developed based on slide-ring friction is intriguing. A crucial concept in the current manuscript is the use of the fast dynamics of slide rings to achieve rapid damping and recovery. However, comparing the developed gels with conventional slide-ring gels does not clarify this point. Conventional slide-ring gels exhibit superior mechanical properties (being highly resilient and stretchable) because the load along the polymer chain can be redistributed due to the movement of slide rings. These gels lack an intentionally introduced damping mechanism. The authors might compare their design to other damping mechanisms, such as physical bonds, to highlight their design's advantages.

Response: We appreciate the reviewer's comments. Following their suggestions, we have conducted a comparative analysis between our friction-based damping design and other mechanisms that involve physical bonds. Typically, damping effects in hydrogels are attributed to viscoelastic dissipation mechanisms, which incorporate sacrificial physical bonds formed through various interactions such as hydrophobic interactions, ionic interactions, hydrogen bonding, coordination interactions, and host-guest interactions. The reformation of these physical bonds is influenced by the rebinding kinetics and the entropic recoiling of polymer strands, processes that are generally slow. This delay hampers the rapid restoration of energy dissipation capacity, especially under multiple load/unload cycles.

In contrast, our design leverages the friction of chain walkers within the polymer network, bypassing the need for the rupture and reformation of physical bonds and avoiding alterations to the hydrogel network structure. We have shown that the fast repositioning of chain walkers via diffusion quickly restores the hydrogel's energy dissipation capabilities within seconds, ensuring sustained damping effects during repeated load/unload cycles. These updates have been incorporated into the revised manuscript (see lines 447-459 on Page 23 in the revised manuscript).

Revisions:

Typically, damping effects in hydrogels are attributed to viscoelastic dissipation mechanisms, which incorporate sacrificial physical bonds formed through various interactions such as hydrophobic interactions, ionic interactions, hydrogen bonding, coordination interactions, and host-guest interactions.⁷⁸⁻⁸³ The reformation of these physical bonds is influenced by the rebinding kinetics and the entropic recoiling of polymer strands, processes that are generally slow. This delay hampers the rapid restoration of energy dissipation capacity, especially under multiple load/unload cycles. In contrast, our design leverages the friction of chain walkers within the polymer network, bypassing the need for the rupture and reformation of physical bonds and avoiding alterations to the hydrogel network structure. We have shown that the fast repositioning of chain walkers via diffusion quickly restores the hydrogel's energy dissipation capabilities within seconds, ensuring sustained damping effects during repeated load/unload cycles.

2. It remains unclear why CD recovery is important for conventional slide-ring gels, as CD recovery is not crucial for achieving their exceptional mechanical properties.

Response: We thank the reviewer for the comments. We apologize for any confusion stemming from our earlier explanation concerning CD recovery in conventional slide-ring gels. In conventional slide-ring gels, the CD rings act as pulleys to cooperatively equalize the tension across polymer chains, positioning themselves where the tension

is balanced on both sides after the initial pulling cycle. This repositioning of CD rings contributes to the exceptional mechanical properties of these gels. As a result, the CD rings do not slide after the first cycle, and the hydrogels exhibit minimal energy dissipation thereafter. Indeed, we concur with the reviewer that the recovery of CD positions is not essential for the exceptional mechanical properties observed in conventional slide-ring gels. However, in our hydrogels, the restoration of CD positions is crucial for maintaining damping effects during cyclic loading. We have clarified this point in the revised manuscript. (See line 463-467 on Page 24 in the revised manuscript)

Revisions:

Additionally, the predefined range of walker movement, set by covalent crosslinks, is critical for rapid recovery of CD positions, which facilitates the fast restoration of the damping capacity in our design. These features distinguish our hydrogels from traditional slide-ring hydrogels, where CD recovery is not crucial for achieving their exceptional mechanical properties.

3. The authors revised Figure 1 to show all slide-ring linkers bridging two network chains. However, the formation of intra-chain slide-ring linkers during gelation is also possible (original Figure 1d). The authors should discuss the potential impact of intra-chain linkers on mechanical properties.

Response: We thank the reviewer for the comments. We agree with the reviewer that there might be the formation of intra-chain slide-ring linkers during gelation. In such instances, the chain walkers (CD-PEG-CD) may not effectively slide along the PEG chain under cyclic loadings. Consequently, the presence of these intra-chain slide-ring linkers may not contribute to the friction-based energy dissipation of hydrogels. Now we have added the new comments in the revised manuscript. (See line 141-143 on Page 7 in the revised manuscript)

Revisions:

Noting that two CD rings of a chain walker may attach to the same PEG chain during

gelation and act as the intra-chain slide-ring linker, which would not contribute to the friction-based energy dissipation of hydrogels.

4. Complex modulus is defined as $G^* = G' + iG''$, or $G^* = \sqrt{(G'^2 + G''^2)}$. Not $G^* = \sqrt{(G' + G'')}$.

Response: We thank the reviewer for the comments. We apologize for the error in our previous response regarding the definition of the complex modulus. It was a clerical mistake. In line with the reviewer's comment, the complex modulus should indeed be defined as $G^* = \sqrt{(G'^2 + G''^2)}$.

Reviewer #3 (Remarks to the Author):

The authors have addressed all the reviewers' concerns, and it is recommended for publication in Nature Communications.

Response: We thank the reviewer for the positive comments.

REVIEWERS' COMMENTS

Reviewer #1 (Remarks to the Author):

I have checked their revised manuscript and they have addressed my inquiries. The revised manuscript is likely suitable for publication in Nat. Commun.

Reviewer #2 (Remarks to the Author):

The authors have addressed all the comments. I recommend this manuscript for publication in Nature Communications.